

# The MetVed model: Development and evaluation of emissions from residential wood combustion at high spatio-temporal resolution in Norway

Henrik Grythe[1], Susana Lopez-Aparicio[1], Matthias Vogt[1], Dam Vo Thanh[1], Claudia Hak[1], Anne Karine Halse[1], Paul Hamer[1], and Gabriela Sousa Santos[1]

[1]Norwegian Institute for Air Research (NILU), P.O.Box 100, 2027 KJELLER

*Correspondence to:* H. Grythe (heg@nilu.no)

**Abstract.**

We present here emissions estimated from a newly developed emission model for residential wood combustion (RWC) at high spatial and temporal resolution, which we title the MetVed model. The model estimates hourly emissions resolved on a 250 m grid resolution for several compounds, including particulate matter (PM), black carbon (BC) and polycyclic aromatic

hydrocarbons (PAH) in Norway for a 12 year period. The model uses novel input data and calculation methods that combine databases built with an unprecedented high level of detail and near national coverage. The model establishes wood burning potential at the grid based on the dependencies between variables that influence emissions; i.e., outdoor temperature, number of and type and size of dwellings, type of available heating technologies, distribution of wood-based heating installations and their associated emission factors. RWC activity with a 1 hr temporal profile was produced by combining heating degree day,

and hourly and weekday activity profiles reported by wood consumer in official statistics. This approach results in an improved characterisation of the spatio-temporal distribution of wood use and subsequently of emissions, required for urban air quality assessments. Whereas most variables are calculated based on bottom up approaches on a 250 m spatial grid, the MetVed model is set up to use official wood consumption at county level, and then distributes consumption to individual grids proportional to the physical traits of the residences within it. MetVed combines consumption with official emission factors that makes the

emissions also upward scalable from the 250 m grid to national level.

The MetVed spatial distribution obtained was compared at urban scale to other existing emissions at the same scale. The annual urban emissions, developed according to different spatial proxies, were found to have differences up to order of magnitude. The MetVed total annual $PM_{2.5}$ emissions in the urban domains compare well to emissions adjusted based on concentration measurements. In addition, hourly $PM_{2.5}$ concentrations estimated by an Eulerian dispersion model using MetVed emissions

was compared to measurements at air quality stations. Both hourly daily profiles and the seasonality of $PM_{2.5}$ show a slight overestimation of $PM_{2.5}$ levels. However, a comparison with black carbon from biomass burning and benzo(a)pyrene measurements indicates higher emissions during winter than that obtained by MetVed. The accuracy of urban emissions from RWC relies on the accuracy of the wood consumption (activity data), emission factors and the spatio-temporal distribution. While there are still knowledge gaps regarding emissions, MetVed represents a vast improvement in the spatial and temporal

distribution of RWC.



# 1 Introduction

Wood burning for residential heating emits to the atmosphere primary aerosol particles, short lived climate gases, and organic volatile and semi-volatile compounds (VOCs, SVOCs), which can condense on existing primary particles, which in turn leads to increased particulate matter mass (e.g. Seljeskog et al., 2017). These aerosol particles play an important role in air quality
and hence on human health. Throughout most urban areas, natural aerosol concentrations are augmented by anthropogenic emission sources, such as particles emitted from traffic and residential heating. Combined with strong emission sources, the conditions in urban areas often prevent efficient dilution of atmospheric pollutants, making citizens disproportionately exposed to high local pollution levels. Together with nitrogen oxides (NOx), elevated particulate matter (PM) concentrations remain a major concern for human health. Especially fine PM from combustion sources are consistently associated with cardiovascular
diseases and mortality (e.g. Pope et al., 2000).

The emissions from residential wood combustion (RWC) are considered a main contributor to harmful atmospheric pollutants in many European cities. For aerosol particles, the mass contribution from RWC is highlighted as a large source across Europe. In Nordic countries, RWC is an especially large source of aerosols, contributing more than 50% of the total urban and national $PM_{2.5}$ anthropogenic emissions, making it the single largest source. RWC have ben found to account for as much
as 50-80% of urban $PM_{2.5}$ mass concentrations (Krecl et al., 2008). In Nordic countries, there is a strong tradition tied to wood burning. The combination of readily available wood supply with an especially strong aesthetic appeal of wood burning stoves (Levander and Bodin , 2014) for the Nordic population leads to many residential buildings relying in part on heating by wood burning during the extended winter period (Denby , 2009). This tradition is widespread across the Nordic area, but there are also some important differences among the Nordic countries. Boilers are commonly used in Sweden and Denmark,
whereas masonry and sauna stoves are more common in Finland (ACAP, 2014). In Norway, the installations for RWC mainly consist of stoves and open fireplaces, which are predominantly small space heaters, conversely to boilers which are typically larger heating devices. This is manifested through an estimated 2.1 million domestic wood burning heating installations in the 2.4 million Norwegian households, with an additional 900.000 in the 1 million cabins and summer houses of Norway (Norsk Varme priv. comm.).

Emissions from RWC are dependent on many factors such as the type and size of the wood (e.g., logs, chips, pellets), the burning efficiency of the wood installation, the draft conditions, fuel load, burning conditions, the moisture content of the wood, and the operation itself (e.g. automatic or manual feeding ACAP, 2014). The resulting outdoor concentration of pollutants from RWC emissions are in addition dependent on the emission altitude, atmospheric conditions and removal efficiency (deposition and dilution). As high RWC activity is often combined with low temperatures, when the frequent presence of temperature
inversions enhances pollution levels, meteorological conditions also play an important role. A temperature inversion occurs when the air temperature rises with altitude resulting in reduced buoyancy of air masses at the surface and consequently the vertical mixing of air is reduced. Since RWC is generally produced within the boundary layer, temperature inversions trap the pollution at low altitudes leading to increased concentration levels.



In Norway, where there are approximately 3 million individual wood burning installations, and so establishing the emissions from each individual point source constitutes a challenge. Besides, emissions from RWC largely differ both temporally and spatially within a city and across regions. It is therefore essential to develop accurate emission inventories, with a high level of both spatial and temporal resolution, that capture both modes of variability. This will support the understanding of the processes

that lead to high pollution episodes in winter, predict them, assess the potential impact on human health and evaluate measures to reduce RWC emissions.

The Norwegian emissions for RWC reported to the Convention on Long-Range Transboundary Air Pollution (CLRTAP; (http://www.ceip.at/)) are calculated based on wood consumption at a county level, derived from self-reporting questionnaires, and information about the available technology, distinguishing between open fireplaces, and wood stoves produced before

1998 and after 1998 (Aasestad et al., 2010). Wood consumption per technology is thereafter combined with official Norwegian emission factors that represent real-world firing conditions (Seljeskog et al., 2013). Different emission factors are used in and around the three largest urban centres in Norway, based on the assumption of different firing habits in urban areas from those used for the rest of the country (NEA, 2018).

There exist several methods to allocate and grid emissions from RWC down to the urban scale. Common to all these RWC

emission methodologies is that they use a downscaling approach to try to resolve households differentiated emissions. To enable this, accurate and detailed input data down to the urban scale is required. Data availability determines to a large extent the type of proxy that can be used to spatially distribute emissions for RWC at high resolution. Thus, the initial and crucial step in the development of high resolution RWC emission inventories is the collection of suitable data.

For the spatial distribution and gridding of emissions from RWC, the most common method is a downscaling approach

applied to existing lower resolution emissions or activity data by means of auxiliary data (Timmermans et al., 2013; GAINS, 2009). The most common parameters used for downscaling have been population or dwelling density. The underlying assumption is that emissions are equal from all households and therefore it positively correlates with the increase of population or dwelling density. Across European countries, this assumption has lead to overestimation of emissions in urban areas (Timmermans et al., 2013). In southern European countries, RWC is more common in rural than in urban areas and therefore emissions

per household decrease with increases in population density (e.g. Terrenoire et al., 2015). In Nordic countries, on the contrary, RWC is common also in urban areas and new proxies have been developed to capture the variability at the urban scale. In Denmark, a differential distribution of emissions according to the type of dwelling is proposed, and accordingly detached houses are assumed to have higher activity than, for instance, apartments (Plejdrup et al., 2016). In Sweden, national total emissions are calculated based on the domestic energy budget calculations and emission factors, then the gridding of emissions is done

based on the number of wood boilers, wood stoves, pellet boilers and oil boilers on a municipality level (Andersson et al., 2015).

Uncertainties in emissions for RWC at the urban scale are due to the activity data (i.e., wood consumption), emission factors per technology and on the spatial-temporal distribution of emissions. In this work, we aim to contribute to the improvement of the distribution of RWC emissions at high spatial and temporal resolutions, and subsequently aim to reduce the uncertain-

ties on emissions and dispersion modelling at the urban scale. Hereby, we describe the MetVed model based on defining the



wood burning potential at 250 m resolution resulting from the analysis of several combined databases built with an unprece-
dented high level of detail. Near national coverage of the amount and types of dwellings, the type of available residential
heating technologies (e.g. district heating, heat pump) and individual wood stove appliances set up the basis of the model.
Emissions are distributed in time based on heating demand which is based on the outdoor temperature. In addition, the model

distributes emissions in two vertical layers depending on whether wood consumption occurs in houses or apartments resulting
in a lower or upper injection layer, respectively. The MetVed model constitutes a significant step forward in the development
and improvement of high resolution emission from RWC.

## 2 Model input data

The MetVed model is set up with several routines for the calculation of wood consumption at the at 250 m grid level. Emission

factors for three different burning technologies are provided by Seljeskog et al. (2013), old stoves (pre-dates 1998), new stoves
(1998) and open fireplaces. Emission factors are combined with aggregated consumption statistics at the county level with the
same differentiation. The other main input data-sets are the location of fireplaces from Fire and Rescue Agencies Registry,
data on dwelling types and available residential heating technology obtained from the largest real estate advertisement portal
in Norway and energy consumption from the Norwegian Energy Labelling System (ENOVA). The model additionally uses

outdoor temperature together with a diurnal and weekly variation of woodfuel consumption in Norway to establish the time
dependence of emissions. As with most emission models, the accuracy and level of detail depends on the available input data.
For MetVed, the model output resolution is determined by the resolution of the input dwelling information (i.e., 250 m).

As the above data-sets constitute the basis for the analysis of RWC in Norwegian households we provide a detailed de-
scription of each dataset in this section. The utilisation of often publicly available high resolution data makes the principles

behind the model versatile and transferable to other countries or emission sectors where similar underlying data can be readily
gathered and applied.

### 2.1 Wood consumption

The use of wood for residential heating for the years from 2005 to 2016 is provided based on the responses to the Statistics
Norway's Travel and Holiday Survey. Data on wood consumption is collected and officially reported by Statistics Norway

at the county level. The survey gathers data in four quarterly surveys covering the preceding twelve months. The calculated
consumption of wood is the average of five consecutive quarterly surveys. The survey contains 25 questions regarding wood
burning for residential heating. The sampling pool of the survey is drawn at a nationwide level and is considered representative
for all counties. Wood consumption in each of the three technology classes (open fireplace, stove produced before 1998 and
stove produced after 1998) is available for each county, and this data is used in the MetVed model as input because finer

resolution is not available.

Reported wood mass (kg) from questionnaires is recalculated to represent dry wood consumption by assuming an 18% water
content. In that way, wood consumption corresponds with emission factors that represent grams of pollutant per kilogram of



dried wood. According to Statistics Norway, there are several elements of uncertainty in the wood consumption data, such as the survey sample size and the employed conversion factors (e.g., mass of bags of wood reported as volume). Statistics Norway concluded that the coefficient of variation of the total wood consumption is below 3% based on an uncertainty study carried out in 2011 (SSB , 2018). The uncertainty is higher at county level as the values are based on smaller samples.

## 2.2 Emission Factors

The Norwegian emission factors are listed in Tab.1 and were established by Seljeskog et al. (2013) for three categories, i) open fireplaces, ii) stoves produced before 1998 and iii) stoves produced after 1998. The emissions factors are determined by laboratory experiments following the Norwegian Standard for testing enclosed wood heaters and smoke emissions (Norsk Standard 3059, 1994) and are considered as representative of "real-world" conditions. The particle sampling is carried out in a dilution tunnel in order to mimic the dilution and cooling effects when the smoke exits the chimney. In this way, the particle sampling also accounts for condensed matter.

Seljeskog et al. (2013) proposed two sets of emission factors depending on different firing condition in Norwegian cities and rural areas. The difference between the two emission factors is that wood stove users in the countryside fill up their stoves to the maximum and then close the combustion air to achieve heating during the night time, whereas in large cities lower part load of the stove is assumed (Haakonsen and Kvingedal, 2001). To our knowledge, there is not a solid and updated study that supports this assumption, thus we presume the same firing conditions as in large cities across all of Norway.

## 2.3 Dwelling number

The data-set containing the type and number of dwellings is obtained from Statistics Norway (Bloch, 2018). The data-set originates from the state tax agency registry (SERG), and that covers all households in Norway. It can be considered complete and up to date. The gridded version at 250 m of this registry maps the number of dwellings, the number of detached houses, duplexes, townhouses, the number of dwellings in apartment blocks, and other dwellings. An example of the dwelling number data-set (for the municipality of Stavanger) is shown in Fig. 1. The grid number of total houses (Fig. 1 a) has the highest density of dwellings in downtown Stavanger. This is also where apartments (Fig. 1 c) have their highest density while detached houses (Fig. 1 b) are much more uniformly spread through out the municipality. This illustrates a typical feature of most urban areas, where a city centre typically consists of mainly apartment blocks, whereas detached houses and duplexes are more prevalent in sub-urban areas. In Norway in 2017, there were around 2.5 million registered dwellings of which 50% are detached houses, 9% duplexes, 12% townhouses, 24% apartments and 5% are classified as others. The spatial distribution of dwelling types differs across region and area. This is especially so for the apartment share, which ranges from 77% in Oslo municipality, to 0% in many rural municipalities.



## 2.4 Fire and Rescue Agencies Registry

In Norway, there are 620 fire stations divided among $\sim 300$ fire department agencies. The fire and rescue agencies are responsible for inspecting and assessing all firing installations. These are dominated by residential heating installations but the database also contains a small number of cooking appliances. The agencies carry out routine inspections as part of fire hazard safety

procedures. During these inspections, information such as geographical location and type of installation is collected in a local database by each agency.

In this study, we contacted 270 fire and rescue agencies in Norway to gain access to their data-set on firing installations. We successfully obtained the complete information from a total of 101 municipalities, covering almost half (1 million) of Norwegian residencies, including the 5 largest cities. The data-set provides detailed information on all residential addresses

inspected, including the geographical location of the pipe, the installations type, technology and model, and whether the technology can be classified as a clean burning technology (i.e. true / false).

The fire and rescue agency registry, if complete, should include all firing installations including those in residential households. There are however a few caveats to these data. Inspections are carried out continuously and over time, and the data obtained generally did not indicate when the inspection was done and some data could be obsolete. There is also a lack of

uniform sampling method among the municipalities, and the type of data supplied was different for each fire departments system. For instance, there were 795 different housing types in the total fire agency data, which were filtered to each of the residential types of buildings and others. Classification of the installation ranged from a general description, to a complete brand and model type, in all consisting of 890 different descriptions. Each one was placed in one of the three categories of wood burning installation or classified as non-wood based. It also appears that the coverage is not fully complete and must

be considered partial even in the municipality where all fire agency data was provided. For example, the fire agency in Oslo municipality informed us of continual dodging of inspections by house owners for various reasons, which results in an estimated 10% unregistered firing installations (Oslo firedep. priv. comm.). As a result, the data are used as a statistical input to the MetVed model. Cross referencing wood consumption and firing installation statistics shows that there is a relatively good agreement with other existing data, and therefore it could also be used directly in the model. Fig. 1 d) shows all the inspected

installations in Stavanger municipality with a density heat map Fig. 1 e). The highest density of wood burning installations is obtained in Stavanger city centre, but there are also other less densely populated areas where fireplace frequency per dwelling is much higher (e.g., Testa, which is North-West of the city centre).

## 2.5 Webcrawled database

The webcrawled dataset is derived from the webcrawling program GoodOvening that scrapes data in a systematic way from

a real state advertisement portal satisfying certain search criteria. It structures data obtained for further data analysis (for more details see Lopez-Aparicio et al., 2018). The webcrawled data-set is continually updated and consisted of 444.000 geopositioned data points within Norway at the time the data was extracted for the MetVed model. Along with the geographical location, the webcrawling extracts data on the characteristics of each dwelling such as the type that they belong to (i.e. detached





house, townhouse, duplex, apartment, etc), the size and the type of the available energy system for residential heating (i.e., wood burning installation, district heating, heat pump, etc). The webcrawling database is aggregated to the 250 m grid resolution and it allows us to establish, with a high level of detail, grids with low or high RWC potential. A zero RWC potential is assigned to those grids with dwellings with no wood-based heating installations. The highest potential would be to those grids with

dwellings where wood-based installation is the sole residential heating source. Between these, the potential is determined as a proportion of wood-based installation to other residential heating technologies.

Fig. 1 f) shows the proportion of wood-based installations for residential heating relative to other listed technologies (e.g., district heating, heat pump, electricity) at 250 m resolution. The city centre shows a low proportion of wood-based technologies (i.e., from 0% to 30%) even though the same area shows the highest number of residential dwellings (Fig. 1, a) and the highest

density of wood-based technologies (Fig. 1, b). The highest proportion of wood-based technologies is observed spread out in the areas outside the city centre. We see similar spatial patterns at the grid level for the proportion of wood-based heating technologies in and around other Norwegian cities.

## 2.6 Database from the Norwegian Energy Labelling System for Dwellings (ENOVA)

The ENOVA energy labelling system was implemented in Norway in July 2010, and is a self assessment report performed by

owners of dwellings and buildings, or qualified experts in the case of new buildings. The ENOVA dataset consists of about $\sim 650.000$ entries that include i) the size of the dwelling; ii) building type (e.g., apartment, small house, office building); iii) building year; iv) energy consumption of the dwelling (kWh); v) primary and secondary heating installations; vi) energy consumption per fuel; vii) geographical information. While individual buildings are listed in the data, the data accessed were used on a municipal and postal code resolution due to data privacy. The information from these data supplement the two

preceding data-sets and also provides a relation between energy consumption for total heating, RWC and building size/type. Wood consumption input statistics to MetVed were used on the county level to correspond with wood consumption data.

## 2.7 Outdoor Temperature

Outdoor temperature is an important controlling factor for RWC because the main purpose of most RWC is the heating supply. The MetVed model uses observed temperatures as input meteorological data. Observed daily mean temperatures were obtained

for the years from 2005 to 2016 for 57 official Norwegian temperature measurement stations through the eklima database (eklima, 2018) to form the basis for time variation of emissions for MetVed. For now we have ensured that urban areas have an observation point within them.

## 2.8 Other data

In addition to geospatial information of Norway, such as municipal and county administrative borders, we have ancillary data

responses to questionnaires on wood consumption in Akershus/Oslo and Sarpsborg/Fredrikstad (Lopez-Aparicio et al., 2017). Together with all of the individual answers of respondents to the questionnaires of the Statistics Norway Holiday Survey



(where wood consumption is reported) they form the basis of establishing assumptions on variability between primary and supplementary wood heating habits.

To assess emissions, dispersion modelling has been used and the results have been compared with observations. To do this, measurement data of $PM_{2.5}$ was taken from the Norwegian Air Quality network monitoring stations that report to the European

air quality database (https://www.eea.europa.eu). PM is measured by applying methods equivalent to the reference method (i.e., TEOM 1400A and Grimm-EDM180) that continuously monitor and log with a time resolution of 1 hour. Measurements of BC in Oslo were obtained at two road side monitoring stations in the winters 2014/2015 and 2015/2016. The measurements were performed with an Aethalometer (Magee Scientific) that measured aerosol light absorption at seven specific wavelengths at 1 min resolution (Hak, 2017). The results at the specific wavelengths are used to determine the contribution from traffic

and wood burning to BC concentration based on the model established by (Sandradewi et al., 2008). Benzo(a)Pyrene (BaP) measurements were taken from the Norwegian monitoring network through active air sampling. The monthly B(a)P values are derived from the analysis of particle filters, which are collected with a 3 day frequency. The identification and quantification of B(a)P is carried out by GC/LRMS.

## 3   The MetVed Model

For the MetVed model MatLab was chosen since there is a wide variety in the type form and file format of the input data, and it allows for easy reading, visualisation and inspection of data flows in the model. The MetVed model contains different routines to estimate emissions at the 250 m grid. In order to calculate emissions, MetVed first calculate gridded wood consumption and emission factors. Spatial and temporal distribution of emissions further require information on location, type and activity of wood burning installations. These are all derived from the input data, which have different scales and resolutions (e.g.,

county, municipality, grid, point). MetVed calculations take into account the physical properties of households and their heating systems, but they do not account for most human behavioural differences. The main calculation of MetVed is to pre-process the input data to arrive at gridded emissions, calculated as:

$$E(c, yr) = C(c, yr) \times EF(c, yr) \tag{1}$$

Where $E(c, yr)$ is emissions $(g\,y^{-1})$ in a grid, $c$, for the year, $yr$, and $C$ and $EF$ are the wood consumption (kg) and emission

factor $(g\,kg^{-1})$, respectively. Though several calculation methods are available in MetVed, the main method to calculate grid EF's and consumption is based on consumption per technology at the county scale. The initial step is thus to calculate a consumption weighted average $EF$ for year of interest, $yr$:

$$EF(yr) = \frac{EF_{new}C_{new,y} + EF_{old}C_{old,yr} + EF_{open}C_{open,yr}}{C_{total,yr}} \tag{2}$$

Where, $EF_{old}$, $EF_{new}$, and $EF_{open}$ refer to the emission factors for old stoves, new stoves and open fireplaces, respectively

(Tab.1), and similarly $C_{new,y}$, $C_{old,y}$ and $C_{open,y}$ refers to the consumption in the different technology classes.





To distribute consumption to each grid, MetVed calculates a wood burning potential ($WP$) in a hierarchical fashion up from dwelling to grid to municipal level and finally up to county, to match the consumption input data resolution. The $WP$ is calculated to distribute in a consistent way the available consumption at the county level, and this establishes the share of total consumption assigned to each grid. The $WP$ is dependent on statistical and physical properties of each residence (e.g., the type and the size), and has two components:

$$WP = W_{HT} \times P_{HT} \tag{3}$$

a consumption weight ($W_{HT}$) and a frequency of wood-based installation for residential heating ($P_{HT}$), calculated per dwelling type ($HT$) in each county. $P_{HT}$ is calculated based on the fraction of dwellings of each type in each county that have listed wood burning installations in the webcrawled data-set. For the consumption weight $W_{HT}$, a linear dependence between energy consumption and dwelling size is applied for each $HT$ and is specific per county. The linear dependency is established from the ENOVA data-set, and the dwelling size applied is the $HT$ average size determined from the webcrawled data-set per county.

Survey data on wood consumption, however, indicate very skewed consumption statistics that are not explained by total heating demand of a dwelling. According to the surveys, a few high intensity users burn up to 30-50 times the average amount of wood consumed, which can amount to more than 10% of the wood consumed in a given county. This differential usage introduces uncertainty and it will also affect consumption. In broad terms we can distribute the usage rate into three categories, i) inactive (existing installations not in use), ii) secondary usage, used as supplementary heating, and iii) primary heating source. At the moment, there is no way to establish exactly which of the listed installations are in disuse. Therefore, we only consider two categories, i.e., installations for secondary usage and as primary heating source. As an example, Bergen fire department estimated that roughly 15% of the wood burning installations are inactive, 70% is sporadically used for social or secondary heating, and the remaining 15% are primary heating sources. These usage shares will have geographical variations because climate, wood availability and prices of heating vary across the country. It is therefore necessary to improve the information on installation usage. This is done by the analysis of webcrawling data-set (Lopez-Aparicio et al., 2018). This data-set provides good statistics on the relative occurrence of both wood burning installations and other heating technologies at high geographical resolution. When it is established that other heating technologies are available (e.g., district heating, heat pump), wood burning installations are assumed to be used as a secondary heating source.

The difference in wood consumption between a RWC installation that is used as a primary heating source and another that is used as a secondary source has been established through the ancillary data in questionnaires (Lopez-Aparicio et al., 2017). We define the ratio (R) between wood consumption for households with primary ($h_1$) and secondary ($h_2$) RWC heating as $R = \frac{h_2}{h_1}$. To represent $h_1$, we used the $15^{th}$ percentile of consumption and for $h_2$ the average wood consumption to get a conservative estimate of R of 4. In the questionnaire data, RWC as primary heating source occurs only in detached houses and therefore this is only applied to this type of dwellings, where 10% of detached houses are assumed to use wood as their primary heating source.

We establish the ratio of wood burning installations to dwellings as $\text{PF}_{HT} = \frac{NFP_{HT}}{N_{HT}}$, where $N_{HT}$ the number of dwellings of a specific type and $NFP_{HT}$ is the number of installations per $HT$. The ratio predicted by webcrawling and rescue and





fire agencies data-sets predicts roughly 1.7 million installations while by an existing national survey is about 1.9 houses with wood based installations (Norstat 2016 survey, Norsk Varme, priv. comm.). Therefore, an adjustment was done to the detached houses to assign enough total installations to residences based on webcrawling that gave about 10% fewer total installations in residential buildings than reported. The probability of having an installation for RWC ($P_{HT}$) is then calculated:

$$P_{HT} = PF_{HT} + PP_{HT} \times R \tag{4}$$

where $PP$ is the fraction of primary heating for each type of dwelling ($HT$). Finally, the consumption in each grid is calculated by distributing the municipal consumption $C_m$ to each grid depending on the number of dwellings of each type ($N_{HT}$) at the grid, $c$, and their associated $WP_{HT}$:

$$C_c = \frac{C_m}{\sum_{m,HT} N_{m,HT} \times WP_{HT}} \times \sum_{c,HT} N_{c,HT} \times WP_{HT} \tag{5}$$

where the $C_m$ is calculated in the same way as $C_c$ in Eq. 5 based on wood consumption at the county. For each year, emissions ($E(c,yr)$) are finally calculated at the grid by multiplying (Eq. 5) and (Eq. 2) at the grid.

### 3.0.1 Time variation

From an annual baseline consumption, MetVed estimates the hourly distribution of wood usage through a calendar year. The time variation of RWC activity is defined to be dependent on the heating demand defined by the outdoor temperature. Therefore the heating degree day (HDD) concept is used. MetVed looks up the geographically nearest meteorological station to obtain the outdoor temperature data-set. Thereafter the HDD is calculated as:

$$HDD = max(0, T_{Threshold} - T) \tag{6}$$

where T is the outdoor daily average temperature in degree Celsius. Regarding $T_{Threshold}$, we follow Stohl et al. (2013) and use $15^oC$ ($HDD_{15}$). Hourly resolution is obtained by coupling the daily consumption to the diurnal heating cycle in Norway. The time of day when RWC occurs is obtained by a survey covering dwellings that provides information regarding the use of the RWC installations both during the day and week (Aasestad et al., 2010). The obtained weekly and hourly activity shows a strong diurnal cycle, where RWC is higher after 17:00. Hourly consumption is obtained by multiplying the daily HDD derived consumption with the hourly average reported activity, which is also weekday dependent. There is higher activity during weekends than weekdays (Haakonsen and Kvingedal, 2001). The consumption differences between weekdays and weekends are smaller than what is suggested by (e.g Finstad et al., 2004; Krecl et al., 2008). However, the diurnal pattern of concentrations they found for RWC is similar to the emissions profile derived based on the activity data, which assumes an emission profile that take into account emissions of the later stages of the firing cycle (e.g. Heringa et al., 2012; Sciare et al., 2013). Fixed public holidays and Easter are treated as weekends.



## 4 Emission Results and Discussion

### 4.1 Time evolution of wood consumption in Norway

The primary model output from MetVed is gridded emissions for Norway in the period of 2005-2016. Fig. 2 shows the spatial distributions of emissions from RWC obtained with MetVed at a 250 m grid resolution in southern Norway and seven domains selected for the evaluation of urban emissions. The distribution of RWC emissions on the 250 m grid are concentrated where there are residential buildings. This is in cities, valleys and along the coast, and consequently they cover a small proportion of the surface area in Norway. Each grid cell has emissions relative to their proportion of the wood burning determined by the number and type of dwellings and available residential heating technology within it. In the urban domains, the lowest emissions (i.e., in the range 0 - 0.14 $\mathrm{t\,y^{-1}\,grid^{-1}}$) are obtained in the outskirts of the urban areas (blue grids in Fig. 2), whereas the highest emissions (i.e., 0.30 - 1.50 $\mathrm{t\,y^{-1}\,grid^{-1}}$ are centred in the urban areas. The internal distribution within each urban domain varies among the cities, as it depends on the distribution characteristics of apartments and houses, and the availability of non wood based residential heating technologies.

Total annual MetVed emissions are closely linked to the total emissions in Norway reported to CLRTAP (Fig. 3 a) as both are estimated using the same wood consumption data and emission factors. MetVed wood consumption is only for residential heating, and the difference with emissions reported to CLRTAP is that the latter includes consumption in cabins. The peak year of wood consumption is 2010, coinciding with an especially cold winter across Norway. Since 2010, consumption has gone down every year except in 2012. Before 2005, there has been a general decline in reported emissions. This is mainly associated with a reduction in wood consumption and an increase in the share of newer technology ovens. From 2005 until 2016, consumption in open fireplaces is relatively constant, varying between 3 and 5%, whereas the share of consumption in "New" stoves has nearly doubled, from 34% in 2005 to 62% in 2016 Fig. 3 b).

### 4.2 Effects of Technological advances

In 1998, regulations were put into place to reduce emission for newly sold stoves, which should not emit more than 12 $\mathrm{g\,kg^{-1}}$ of $\mathrm{PM_{2.5}}$. In Fig. 3 c) the red dashed line is the assumed average EF of stove sold in that year. Assuming ovens have equal usage, the share in a given year predicted by an exchange of stoves by sales is shown as a dashed yellow line. This $EF$ fits with the derived emission factor for $\mathrm{PM_{2.5}}$ from both MetVed and from the CLRTAP emissions since 2005. Similarly for $\mathrm{PM_{2.5}}$, all other compounds (Fig. 3 d) show a general decline in EFs since 2005. This decline in EFs is the main driver for the decreasing trend in emissions in Norway, as an increasing fraction of wood is consumed in new stoves. BC is the only EF that does not decrease uniformly. The reason is that the EF for open fireplaces is an order of magnitude higher than those for stoves, and the slight variability in the consumption estimates in open fireplaces drives the change.

The MetVed input data was analysed to evaluate both the share of stoves and their use. In the fire and rescue agency data, roughly 70% indicate the age of the installation, of which only 34% of residential stoves are noted to be newer than 1998. A recent survey carried out in Norway in 2016 shows around 64% of the wood burning installations in 2016 are new stoves (Norstat 2016 survey, Norsk Varme, priv. comm.). With current sales estimates the transfer to cleaner technologies will





therefore go on for the next several decades. Similarly, the increased efficiency of newer ovens from an estimated 50% to 75% for new ovens (Seljeskog et al., 2013) will act to reduce consumption. Thus, based on this, future emissions are expected to go down further. The red bars in Fig. 3 b) show an assumed share of existing stove technology assuming annual sales of new ovens to the residential sector of 67 500 units per year. This sale is derived from a fit to emission factors and will be influenced

by consumption differences. Based on the Norstat 2016 survey, annual sales are close to 40 000 installations per year (Norsk Varme priv. comm). This difference indicates that there are large consumption differences and that on average a new installation may involve a higher consumption of fuelwood. Therefore, what the real effect of exchanging an old for a new installation will have on the overall consumption, and therefore on emissions, is still uncertain.

It is worth noting that the two stove technologies "New" and "Old" are comprised of an assembly of stoves. Producers of

stoves today claim to have significantly reduced $PM_{2.5}$ emissions even further since 1998, to about $2 \ \mathrm{g \ kg^{-1}}$ in 2016 (Norsk Varme priv. comm). Both CLRTAP and MetVed assumes constant EF for the "New" oven assembly. The dotted lines in Fig. 3 c) show the EF with a continuing reduction down to $2 \ \mathrm{g \ kg^{-1}}$ in 2016, both for ovens sold in that year (light blue) and for the share of stoves in that year (green). Were EF based on this it would act to significantly reduce 2016 $PM_{2.5}$ EF from 13.5 to 7.4 $\mathrm{g.kg^{-1}}$, and thus nearly halving emissions.

### 4.3  Temperature dependence of wood consumption

Comsumption of fuelwood follows the change in demand for heating energy. In the Fig. 4 a) the total annual number for $HDD_{15}$ at observation sites (57 Meteorological stations) across the simulation domains in Norway are put together weighted by the total number of dwellings in the domain covered by each station. The MetVed annual consumption is from the CLRTAP (Fig. 3 a), which is derived independently from the temperature. The linear fit to the period 2005 to 2016 indicates that

consumption is reduced by ∼6% per year whereas a slight increase in HDD demand is observed over the same period. Residuals from both trends show that the temperature can explain about 63% of the variance in consumption (r = 0.79 Fig. 4 b). HDD would therefore provide a good indicator for present and future annual consumption variations, but long term trends depending on the physical properties of residences must also be considered. It is also important to note that the trend of decreasing consumption between 2005 and 2016 is not due to increasing average winter temperatures.

In MetVed model, the concept of HDD is only used to distribute wood consumption within a year, and only to days with a heating demand. The model determines the total number of HDD in a year over which the consumption is distributed. The choice of a threshold HDDs, i.e., the coldest temperature where no heating is required, only influences the temperature sensitivity of consumption within a year. A lower HDD threshold would therefore only lead to higher emissions in winter and less during spring and autumn. In the cold year 2010, consumption increased by 25%, the number of HDDs with threshold 20

degrees ($HDD_{20}$), increased by 23% relative to the 2005-2016 mean. Whilst $HDD_{15}$, $HDD_{10}$ and $HDD_5$ showed an increase



of 36%, 65%, 135%, respectively. Thus a lower threshold temperature will thus increase temperature sensitivity of emissions. From the residual consumption and $HDD_{15}$ in Fig. 4 b) we derived the relation for the period 2005-2016:

$$\Delta C = 0.77 \ \Delta \ HDD_{15} \qquad (7)$$

where $\Delta C$ is the change in consumption, $\Delta HDD_{15}$ the change in the $HDD_{15}$, both unit-less. The relation between the outdoor average temperature and consumption is $0.32 \ kg \ dwelling^{-1} \ HDD_{15}^{-1}$.

As other factors influence wood consumption, household energy consumption per $m^2$ has been derived from the Norwegian energy balance model (https://www.ssb.no/energi-og-industri/statistikker/husenergi/hvert-3-aar/2014-07-14) and shown in Fig. 4 c. An increase in the number of energy efficient houses and a lower wood energy share reduce the annual wood energy demand (both by $\sim 1\%$ per year in Fig. 4 c). Seljeskog et al. (2013) report an energy efficiency of 15%, 50% and 75% for open fireplaces, old stoves and new stoves, respectively, which are also used in the energy calculations. Consumption per technology in 2012 gives a 61% average stove efficiency, and at a constant energy output. The effect of newer, more efficient installations has influenced the decrease in consumption by 0.5% per year (Wood efficiency in Fig. 4 c).

In 2012, the average reported dwellings in Norway received 16% of their energy, about 3200 kWh, from wood. This is significantly higher than the average energy share suggested by the MetVed input data from ENOVA, where the annual average energy from wood is 960 kWh, although only 3% reported using wood for heating. The reason for the lower reported wood consumption in the ENOVA data-set is not clear, but may be related to conscious under-reporting to achieve a better energy certificate by the dwelling owners. Birch wood, which is the most common fuelwood in Norway, has a dry (0% moisture) energy content of $4.395 \ kWh \ kg^{-1}$ (Raymer, 2006). To achieve the stated energy, an average Norwegian household would then in 2012 have to burn 1195 kg of dry birch wood. The same conversion of the ENOVA reported consumption gives 7.2 kg of dry wood for an average Norwegian household. Based on the official wood consumption data, the total residential consumption in 2012 was 1460 kton which equates to 589 kg of wood consumption per dwelling (800 kg per dwelling with wood based installation). Due to these large discrepancies in the data, the housing type and size and energy dependencies calculated within MetVed done based on the ENOVA reported total energy consumption and not directly on the reported energy consumption based on wood.

## 4.4 $PM_{2.5}$ Emissions at Urban Scale

The main aim of MetVed was to improve the spatial distribution of emissions from RWC. Resulting emissions of $PM_{2.5}$ within each of the urban domains in Fig. 2 from MetVed are shown with four other Norwegian RWC emission inventories in Fig. 5 a). These domains contain the largest cities in Norway and together cover 44% of Norwegian dwellings. With the exception of NBV emissions, all urban emission inventories are obtained by downscaling national emissions submitted to the CLRTAP (http://www.ceip.at/). Though the year of reference varies between inventories, the magnitude of total national emission are comparable. The difference within these domains is therefore to a large extent determined by the method of spatially distributing the emissions.



The EMEP emissions are distributed on a $0.1^o$ or about 7 km grid, and represent the lowest urban emissions in all domains, except Oslo, having an order of magnitude lower emissions than the remaining domains. To our knowledge, detailed information about the method used to downscale the EMEP emissions is not publicly available. NEA (2018) states that when the activity data used to estimate emissions is available at a higher geographical resolution than national, it is used to distribute emissions. In Norway, wood consumption is available at county level, thus we assume that EMEP emissions from RWC are distributed at this level. This is consistent with the visualisation of emissions where the county administrative borders are visible, and emissions are widely distributed in the Norwegian geography, covering also unpopulated areas. The TNO-MACC emissions have the same spatial resolution as those from EMEP (i.e., a ∼7 km grid), but have much higher emissions within each urban domain. TNO-MACC uses internal approaches based on population density and a function to describe proximity to wood as their downscaling method (Kuenen et al., 2014).

NWA emissions (NordicWelfAir, http://projects.au.dk/nordicwelfair/) are on a 1 km grid and are based on scaling down wood consumption per technology from county level based on dwelling number at 250 m resolution. Different RWC activity is assumed for apartments and houses, and the Norwegian official emission factors (Tab.1). Thereafter emissions are re-gridded to a 1 km resolution. NBV emissions, also on a 1 km grid results from downscaling wood consumption per technology by dwelling number at the district level resolution also using the Norwegian official emission factors (Tarrasón et al., 2017). The domain of Oslo in NBV is an exception, in this case emissions are reported in Lopez-Aparicio et al. (2017). In NBV, a multiplication scaling factor was derived for each urban domain individually based on the ratio of concentration levels obtained by atmospheric dispersion modelling to observed $PM_{2.5}$ levels. These factors vary from 0.42 to 0.27, i.e, dividing the total emissions in these areas by a factor (2.3-3.7). For the MetVed emission inventory, the total emissions within the area is dependent on all the input parameters detailed in Section 3.

The total emissions within each domain is for all inventories (with the exception of EMEP) are closely related to the number of houses within each domain (Fig. 5 b), either directly or indirectly through population. Note that the sizes of the domains vary somewhat and all have different total area (Fig. 3), but all cover a city and their area of influence. The dependency on dwelling or population density differs among the methods, and thus emissions at urban scale vary accordingly. TNO-MACC results in the highest emissions at urban scale followed by NWA emissions, but EMEP results in very low emissions at the same scale as the downscaling approach distributes and diffuses emissions at county level. In the MetVed model, the emissions are determined by the physical properties of the houses and their installed heating technologies within each domain (also done on the 250 m grid). In our study, we consider NBV emissions as a tuned emission dataset as they have been obtained by comparing results from dispersion modelling and observations, and then scaling emissions. The correction factors vary among the domains and therefore the methodology lacks consistency. However, MetVed emissions are produced at high resolution at national level following a consistent methodology and relationships between variables that influence emissions. In addition, MetVed emissions at national levels equal the official emissions reported to the CLRTAP which gives it consistency as a national emission inventory. The results from the MetVed model at the urban scale are consistent with NBV for most of the domains. MetVed emissions within the urban domains have, for most of the domains, lower emissions than those derived as a result of downscaling approaches based on dwelling density or number (i.e., TNO-MACC, NWA). The MetVed emissions are the



most similar to the locally corrected NBV emissions. This lowering of urban emissions in MetVed relative to emissions from downscaling approach is the result of taking into account the share of wood-based technologies from the webcrawled input data. In that way, we account for lower wood consumption when heating technologies other than wood-based technologies are available (Plejdrup et al., 2016).

There are a number of dependencies in the consumption and wood installation statistics for these domains (Fig. 5 b). For instance, Trondheim has the 4th highest population, is located in the county with the second highest consumption per dwelling, and it has the second highest consumption per wood installation (after Grenland) of any domain. For NBV emissions, this leads to an observation based emission correction of 0.33. In MetVed, Trondheim's high share of apartments (Fig. 5 c) and low frequency of fireplaces (Fig. 5 b) gives the lowest emissions for all of the MetVed domains, which shows that these properties

act well to explain the reduced consumption. Dwelling size is an important factor for calculating consumption in MetVed, and it plays a significant role because apartments in the Trondheim city centre are generally small. The Oslo domain similarly has low emissions per dwelling, but in this case it is also driven by a low consumption per wood burning installation.

  An increased prevalence of wood installations is positively correlated ($R^2 = 0.44$) with higher consumption within the domains (Fig. 5 c). This is a result of the MetVed calculations, where the combination of regional consumption statistics and

the prevalence of wood installations is found by the webcrawled statistics. This acts to support assumptions made in previous studies, e.g., "proximity to wood" and rural consumption being higher than urban consumption, but is in MetVed represented in dwellings' heating installations and their usage rate. MetVed additionally also considers the dwelling size which further increases a rural amplification of firewood use, as rural properties are on average larger.

## 5 Comparison with observations and evaluation

Atmospheric dispersion modelling was carried out for $PM_{2.5}$ for the year 2015 with supplementary runs for BC and additional investigation of benzo[a]pyrene (B(a)P) concentrations at sites where measurements were available. In the selected Norwegian areas, B(a)P measurements may be considered a tracer for wood burning activity. The black carbon from biomass burning ($BC_{BB}$) measurements available enable comparison with RWC BC concentrations. The evaluation of BC and B(a)P adds to the comparisons to $PM_{2.5}$ measurements, which have the added uncertainty of other sources. The combined outcomes from

these evaluations will add to the understanding of the uncertainties associated with emissions from RWC and their contribution to air pollution levels in urban areas.

  RWC emissions from MetVed for 2015 were used as input in the air quality model EPISODE, an off-line Eulerian dispersion model frequently applied to assess air quality in Norwegian cities. EPISODE is driven by meteorology from AROME model (Seity et al., 2011) at 1 km resolution. $PM_{2.5}$ background concentrations come from CAMS daily forecast re-gridded to the

same vertical and horizontal grid as the EPISODE model (Marécal et al., 2015). In EPISODE, $PM_{2.5}$ is treated as an inert particle subject only to removal by transport, which limits the size of the modelling domains. Furthermore, EPISODE requires emissions on the same scale as the meteorology, therefore MetVed emissions were regridded to the meteorological field grid (1 km) within each domain.



## 5.1 Particulate Matter (PM$_{2.5}$)

The dispersion modelling in the 7 domains of PM$_{2.5}$ was compared to available measurements of total PM$_{2.5}$ concentration. Comparisons must therefore include all anthropogenic emissions in the urban areas. The additional emission inventories include, along with RWC, emissions by shipping, off-road machinery, traffic exhaust and suspension of road dust. Urban emissions are estimated based on high resolution input data, that thereafter are aggregated to a 1 km grid, and combined with time variation functions to result in emissions at $1 \, \text{h} \, \text{km}^2$ resolution (for more details see Tarrasón et al., 2017; Lopez-Aparicio and Vo Thanh , 2017). Non-exhaust emissions associated with tyre and road wearing processes and the suspension of road dust was modelled by the vehicle abrasion and suspension model NORTRIP (Denby et al., 2013). Emission domains and the year of reference (2015) were chosen to be able to compare with modelled concentrations obtained based on NBV emissions (Tarrasón et al., 2017). Similarly, measurement stations were selected to coincide with the existing model concentrations and assess a potential improvement when comparing with previous estimates.

With few exceptions, the air quality stations are classified as traffic stations and are operated by the road authorities. Few measurements are therefore ideally located with regards to a detailed evaluation of the spatial distribution of emissions from RWC within each domain. The main difference of the spatial distribution provided by MetVed and that by other methods not differentiating housing types (and size) and available residential heating technologies can best be seen through comparison between stations situated between different types of buildings. In Oslo for instance, only one monitoring station (Smestad) could be qualified as located in an area dominated by detached houses, together with similarly located monitoring stations Vaaland in Stavanger and Lensmannsdalen in Grenland. The remaining stations are in close proximity to apartments or a diversity of dwelling types, along with roads with large emissions. Together with the contributed uncertainty from other sources the differentiation on housing types is therefore hard to evaluate, and so the main focus is on evaluating total emissions within each domain.

Hourly correlation coefficients and biases are shown in Tab. 2. Comparing to NBV emissions which were calibrated with a weekly correction factor, MetVed temporal emission profile improves correlation at 7 of the 8 stations. The bias is not improved, because the MetVed emissions are generally higher, and so the overestimation by the model is increased. The Oslo comparison shows a strong overestimation, and a possible reason is that the MetVed approach has a limited effect in Oslo, as it is in itself a county and so total consumption and emissions are direct results of the input wood consumption. In the Trondheim domain, the NBV emissions results from applying a factor of 0.33 to the dwelling density derived emissions (Tarrasón et al., 2017), and MetVed produces similar results as NBV at the Elgeseter monitoring station. The consideration of dwelling type and size, and the fireplace information accounted for in MetVed, account similarly for lower emissions and concentrations in urban area as the local scalings applied individually in NBV.

Annually averaged observed concentration at the sites is $8.0 \, \mu\text{gm}^{-3}$ and using MetVed emissions this is $8.7 \, \mu\text{gm}^{-3}$ for the MetVed modelled concentration. Monthly observed concentrations at each station are shown in Fig.6 as a black line, and the modelled concentrations are shown as bars, coloured by sector. Both model and observations show a pronounced seasonal cycle in PM$_{2.5}$ concentrations with a winter peak primarily driven by the background concentration and RWC contributions, and the



remaining sectors have a similar cycle, but contribute less to the $PM_{2.5}$ seasonality. Model concentrations, including RWC, have a winter to summer (i.e., DJF to JJA) ratio of ~2, and the measurements have the winter (DJF) to summer (JJA) ratio at 1.45 indicating that the seasonality in model concentration is more than twice as strong. As wood burning for residential heating is an intensive winter activity in Norway, it is hard to envision that there should be no seasonality in RWC, and

therefore the uncertainty on the seasonality of other contributing sectors must be considered. Yttri et al. (2011) reported, based on simultaneous measurements in Oslo and a regional background site (Hurdal, 70 km NE Oslo), similar $PM_1$ values in summer at both sites, 7.6 and 7.7 µgm$^{-3}$, respectively. However, in winter, $PM_1$ concentrations in Oslo was similar to summer concentrations, i.e., 7.8 µgm$^{-3}$, but at regional background site was 45% lower than in summer, i.e., 4.3 µgm$^{-3}$. Source apportionment also indicated that the elevated urban wintertime concentration consisted largely of organic mass from biomass

burning. These results indicate a strong urban source rather than influence from regional background levels during winter.

The diurnal evening $PM_{2.5}$ peak in winter (Fig. 6, bottom right) is also more pronounced in the model than that in the observations. While the difference between summer and the winter diurnal pattern is also enhanced by meteorological conditions, the absolute difference between winter and summer (i.e., DFJ -JJA gray shape in Fig. 6, bottom right) is much less prominent than the diurnal winter contribution of RWC alone (orange line in Fig. 6, bottom right). The assessment of the diurnal variation

also suggests that the total modelled influence of RWC on the air quality station $PM_{2.5}$ is too strong.

In Oslo, three model simulations were used to assess the sensitivity of surface concentrations to emission altitude. In urban Oslo, the annual average RWC concentration of $PM_{2.5}$ at 2 m was 4.41 µgm$^{-3}$ when all RWC emissions were emitted in the surface layer (0-30 m). When apartment emissions were emitted in the second layer the surface concentration was reduced to 3.76 µgm$^{-3}$, and when smaller buildings emit in the second layer, and this further reduction to 3.19 µgm$^{-3}$ is observed.

The winter time diurnal cycle of $PM_{2.5}$ has a more pronounced peak, and the seasonal cycle of $PM_{2.5}$ similarly indicates a too strong seasonal cycle obtained with all emission estimates, when compared to observations and so the total contribution of RWC to $PM_{2.5}$ concentrations seems to be overall overestimated. This is supported by the new emission inventory developed by van der Gon et al. (2015) for particulate emissions from RWC. When assessing organic carbon emissions, van der Gon et al. (2015) establish that while emissions in most of Europe are underestimated, in Norway they are overestimated.

**5.2  Black Carbon**

During both winter 2014-2015 and 2015-2016, hourly concentrations of aerosol absorption were measured at two sites in Oslo (Smestad and RV4 Fig. 7). The first period was at Smestad (17.Dec.2014-18.Mar.2015) and the second at RV4 (20.Nov.2015-30.May.2016). The decomposition of the absorbing aerosols, includes BC from biomass burning ($BC_{BB}$), which in Oslo (in winter) should be equivalent to BC from RWC. While both measurements were done in close proximity to larger trunk roads,

Smestad is surrounded by a large area of detached houses while the RV4 sampling site is characterised by apartment buildings and a hospital. Even though the number of dwellings in the area surrounding RV4 is higher with nearly 30% more residences within 300 m, MetVed emissions in the grid of RV4 are 10% lower, as the area is characterised by a high share of apartment buildings with low MetVed emissions.

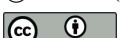



The modelled year (2015) does not completely overlap with either of the two measurements Fig. 7 a), and therefore model data is shown for the temperature range ($T<8^oC$), the temperature range covered by the observations in Fig. 7 b). Daily average $BC_{BB}$ concentrations at both sites show a similar dependence on temperature Fig. 7 c), though overall lower concentration levels are observed in RV4, which is probably influenced by the measurements continuing further into spring. The differ-
ence between the two sites is somewhat smaller in the model. Model concentrations show a similar but weaker temperature dependence, suggestive that daily emissions could increase more than emissions obtained by $HDD_{15}$.

The modelled diurnal variability in concentration (Fig. 7 c) agrees well with $BC_{BB}$. The diurnal time variation of wood consumption and the subsequent MetVed emissions is in agreement with the $BC_{BB}$ observations. The diurnal weekend emission profile is similar to the observed hourly concentration profile, and the total average model concentration fits ($T<8^oC$) well with
the levels of measured $BC_{BB}$.

## 5.3 Benzo(a)Pyrene

Most of the air quality stations, dispersion model results show an overestimation of winter $PM_{2.5}$ concentrations, and a similar underestimation of summer concentrations. While $PM_{2.5}$ have many sources, B(a)P filter measurements offer a way to more directly investigate the RWC contribution.

Fig 8 shows monthly average concentrations between 2015 and 2017 at different urban sites, within the domains in Fig. 2, along with the annual average B(a)P concentration at Birkenes, a regional background air quality observatory. All measurement sites show the same B(a)P seasonality with highest values in January. The most pronounced B(a)P profile is obtained in Lillehammer, followed by Oslo and Drammen, where measurements show similar monthly profiles and concentration levels, and Trondheim, where the lowest B(a)P levels are obtained. The B(a)P annual mean concentration in all urban areas varies
between 0.18, 0.30 and 0.20 $ng\ m^{-3}$ in Trondheim to 0.56, 0.61 and 0.68 $ng\ m^{-3}$ in Lillehammer, in 2015, 2016 and 2017, respectively. These levels are below the B(a)P European target value (i.e., 1 $ng\ m^{-3}$), but above the reference value established by the World Health Organisation (i.e., 0.12 $ng\ m^{-3}$). Unlike, in Birkenes, where values represent regional background levels, B(a)P annual mean concentrations are below the WHO reference value, measured at 0.013, 0.010 and 0.011 $ng\ m^{-3}$, in 2015, 2016 and 2017, respectively. Fig 8 shows in addition the monthly normalised $HDD_{15}$, $HDD_{10}$ and $HDD_5$ obtained
for Oslo (gray shades areas) based on outdoor temperatures for the same period. In MetVed, the monthly normalised HHD relates outdoor temperature with activity (i.e., wood consumption). Fig 8 shows the effect that different thresholds (i.e., 5, 10 or 15 $^oC$) would have on monthly wood consumption, and therefore emissions. A higher temperature threshold will increase activity into spring and autumn, and a lower temperature threshold will limit activity to winter time. The overall B(a)P best fit to these profiles is obtained using $HDD_5$ suggesting a more intense source during winter than what is obtained with $HDD_{15}$.
This is in agreement with the BC observations, opposite of what is observed based on $PM_{2.5}$ measurements.

The annual B(a)P average level at each site is indicative of total RWC activity in the area, and the levels are well in agreement with the spatial distribution of emissions in the MetVed model. Lillehammer, with the highest B(a)P levels, is the least densely populated area of those considered here, but has high emissions from RWC due to high wood consumption in the region and the area surrounding the station is made up of mainly detached houses with a high wood burning potential.





Trondheim, with the lowest B(a)P levels, is a highly populated urban area located in a county with a high wood consumption. The proxies behind MetVed entail the lowest emissions compared with other urban areas (Fig. 5 a) as a result of the high share of apartments with alternative residential heating sources other than wood based represent the observed B(a)P differences well.

## 6   Conclusions

The uncertainties in emissions from RWC at the urban scale rely on those in the activity (wood consumption), emission factors and spatio-temporal distribution of the emissions. With the development of the MetVed model, our aim was to reduce the uncertainties associated with the spatio-temporal distribution of RWC emissions for their use in air quality modelling at urban scale. As the spatial distribution alone cannot explain the large uncertainties, a detailed evaluation of the estimations and evolution of wood consumption and emission factors was also performed. The emissions from RWC in Norway show a significant declining trend. This is driven primarily by the increased use of new technologies for RWC along with a general decline in heating demand from wood based installations through a lower heating demand per $\mathrm{m}^2$ and a lower share of total demand being filled from RWC.

MetVed takes into account the physical properties of residences, and based on the frequency, size and type of dwelling and its available heating technologies, estimates a wood burning potential at the grid level. Even though MetVed takes into account most of the variables that affect emissions from RWC, there are still factors not considered which may affect local emissions, e.g., human behaviour, or specific geo-localised information on which installations are unused. Compared with existing emission inventories, MetVed has a higher spatial resolution, which is supported by detailed input data. The unique set of input data, and the established relationships, lead to an improved horizontal and vertical spatial distribution of emissions. One of the main effects of the MetVed approach, when comparing with other methods, is that emissions are displaced from highly populated downtown areas to the urban outskirts. As a result, MetVed gives lower emissions in highly populated areas than those established by downscaling approaches based on population or housing density, and is moreover in agreement with total local emissions developed by combining results from dispersion modelling and observations. The new approach may have implications when estimating population exposure to pollution levels associated with RWC, as lower population exposure may be expected.

Downscaled emissions derived from annual wood consumption with national emission factors have in the past produced outcomes that did not compare well with observations in Norwegian cities. When comparing with air quality monitoring stations measuring $\mathrm{PM}_{2.5}$, which exist predominantly in urban areas, modelled results have shown a strong tendency to overestimate concentrations. The MetVed model applies several new data sources to the effect of drastically reducing the emissions in urban domains when comparing with direct downscaling methods, and the results are similar to those adjusted by measurement data. The overall correlation is improved even compared to emissions adjusted with observations (NBV emissions) and this highlights the advantages of the MetVed methodology for improving the temporal variability. However, MetVed emissions still overestimate $\mathrm{PM}_{2.5}$ concentrations. The comparison of $\mathrm{BC}_{\mathrm{BB}}$ indicates, for the same domain where $\mathrm{PM}_{2.5}$ is overestimated,



that emissions are on the contrary lower and less temperature dependent than observations should predict. A potential reason for these discrepancies is that $PM_{2.5}$ Norwegian emission factor applied could be too high, which would fit well with all the inter-comparisons with observations. This is also supported by the diurnal cycle concentration profile of MetVed emissions, which have a very similar but stronger profile to that predicted by observations of $PM_{2.5}$ and $BC_{BB}$, which implies that $PM_{2.5}$

emissions are somewhat overestimated.

The temporal emission distribution follows the Heating Degree Day (HDD) combined with a diurnal consumption derived from consumer statistics. The applied HDD improves correlation against measurements relative to the emissions adjusted based on observations. However, this gives more intense emissions during winter, and a stronger diurnal variability compared with profiles inferred from $PM_{2.5}$ observations. The Annual average bias is only 0.23 $\mu gm^{-3}$ (or 3.83%), where the winter

overestimation is compensated by the summer underestimation. Across all stations for which simulations were done, the same general temporal pattern is seen. Substantially higher RWC emissions in the summer months or that emissions occur much earlier in the day across all domains are not plausible reasons for the observed discrepancies between winter and summer. Besides, observed $BC_{BB}$ and $B(a)P$ indicate a stronger dependency on temperature, which would produce a stronger seasonality, than predicted by $HDD_{15}$ used in MetVed. We are confident that the spatial distribution of emissions given by MetVed model

entails less uncertainties than previous methods based on downscaling approaches using population or dwelling number. Thus, further investigation of the accuracy and representativeness of the activity data (wood consumption) and the official Norwegian emission factors is needed. In addition, the main contributor to the seasonality in $PM_{2.5}$ is, along with RWC, the background concentration which does not have a diurnal cycle. The evaluation of the $PM_{2.5}$ seasonality shows the need for improving the time variation of all contributing emitting sectors.

In Oslo, simulations on the vertical distribution of emissions showed that injecting apartment emissions in a higher (30-60m) layer than the other housing types (0-30m layer) altitude, resulted in lower RWC $PM_{2.5}$ surface concentrations in EPISODE by about 2 $\mu gm^{-3}$ (18%). A similar effect at the surface was (further reduction of 14%) observed for moving all emissions up into the model second layer. This shows the sensitivity to emission altitude when comparing with surface concentrations.

Even though the model is developed for Norway, the principles behind it and the methodology can be applicable to other

European countries where similar input data is available. Besides, the MetVed principles, based on data collection and analysis of the dependencies between variables, could be used for other emitting sectors than RWC to improve the spatial distribution of emissions at high resolution. To further improve emission inventories, there is a need for more measurements specifically targeting RWC in Norway, as most measurements are limited to $PM_{2.5}$ at road side stations in urban areas where the signal to noise ratio of RWC is very low. The results and evidence from our study points to even higher emissions from RWC than

than predicted by observations. As MetVed reduced the uncertainties associated with spatio-temporal distribution of emissions, there is a need to revise the activity data and emission factors used for the official reporting of emissions.





## 7 Author contribution

H. Grythe and S. Lopez-Aparicio contributed equally to the text and work within this paper. S. Lopez-Aparicio did all the MetVed EPISODE model simulations. M. Vogt contributed to both text and method & model development. C. Hak provided the BC measurement data and contributed comments and corrections to the manuscript. A. K. Halse provided the B(a)P as well

5 as comments and corrections to the manuscript. D. V. Tranh did data processing and supplied emissions for other sectors. P. Hamer supplied background concentrations and commented on and proofread the manuscript. G. S. Santos did all the EPISODE NBV simulations and provided technical assistance with EPISODE and exiswting emissions.

## 8 Acknowledgement

The model development was funded by the Norwegian Environmental Agency through the MetVed project, and additional fi-

10 nancial support was provided from the Research Council of Norway (iResponse project; 247884/O70), NordForsk (NordicWelfAir Project; #75007) and NILU Institute Strategic Initiative (SIS-MASTER project). We thank ENOVA for sharing their anony-mous data on residences energy consumption, and all 90 fire and rescue agencies that supplied data on firing installations.



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





**Table 1.** Emission factors as amount of pollutant per amount of dry wood (Seljeskog et al., 2013)

| Pollutant | Open Fireplace | Wood Stove (-1998) | Wood Stove (1998-) |
|---|---|---|---|
| $CO$ $(g.kg^{-}1)$ | 126.3 | 150 | 50.5 |
| $CH_4$ $(g.kg^{-}1)$ | 5.3 | 5.3 | 5.3 |
| $PM_{10}$ $(g.kg^{-}1)$ | 17.0 | 17.1 | 12.0 |
| $PM_{2.5}$ $(g.kg^{-}1)$ | 16.4 | 16.5 | 11.6 |
| $BC$ $(\%PM_{2.5})$ | 9 | 1.01 | 0.9 |
| $PAH_{TOTAL}$ $(g.ton^{-}1)$ | 17.4 | 52 | 0.0226 |

**Table 2.** 2015 hourly concentrations fit to observations for EPISODE simulations of $PM_{2.5}$ for each of the stations. The hourly R is the Pearsons correlation coefficient and Bias is calculated as model-observation.

| Station | Domain | R Metved | Bias Metved | R NBV | Bias NBV |
|---|---|---|---|---|---|
| Aakerbergveien | Oslo | 0.58 | 0.63 | 0.56 | -0.57 |
| Hjortnes | Oslo | 0.52 | 1.73 | 0.52 | 0.34 |
| Sofienbergparken | Oslo | 0.35 | 1.21 | 0.33 | -0.24 |
| Danmarks Plass | Bergen | 0.43 | 1.16 | 0.44 | 0.10 |
| Vaaland | Stavanger | 0.40 | 1.44 | 0.33 | 1.36 |
| St.Croix | Nedre Glomma | 0.64 | -1.50 | 0.52 | -1.28 |
| Elgeseter | Trondheim | 0.43 | 2.62 | 0.38 | 2.74 |
| Lensmannsdalen | Grenland | 0.33 | -1.58 | 0.31 | 0.23 |
| Total | | 0.46 | 0.93 | 0.42 | 0.33 |



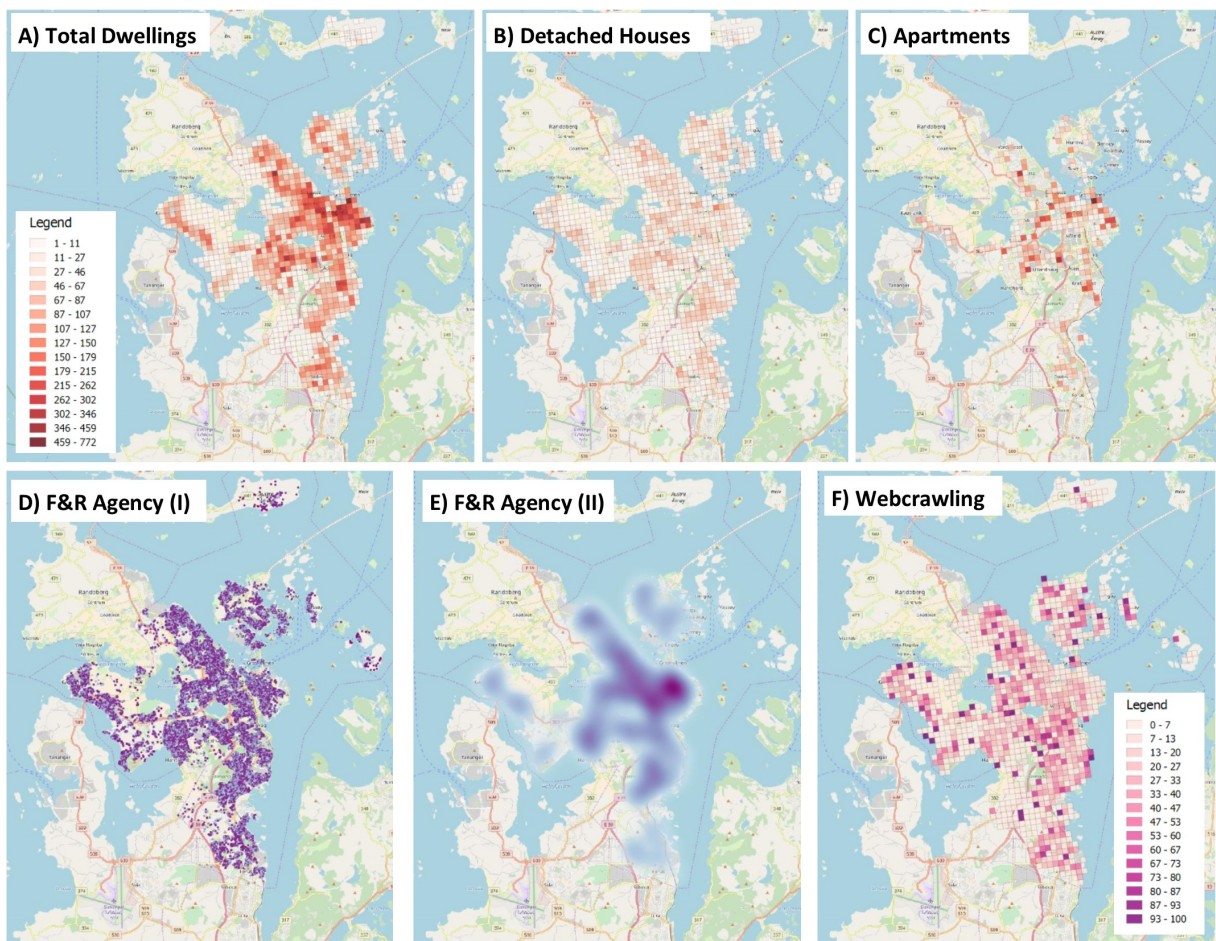

**Figure 1.** Example of part of the input data used in MetVed model in the Municipality of Stavanger. A: Total dwelling number at 250 m grid resolution. B: Number of detached hours. C: Number of apartments. D: Individual wood burning installations from the Fire and Rescue Agency. E: Density of wood burning installations. F: Share (%) of wood based installations for residential heating obtained from the webcrawled data-set.



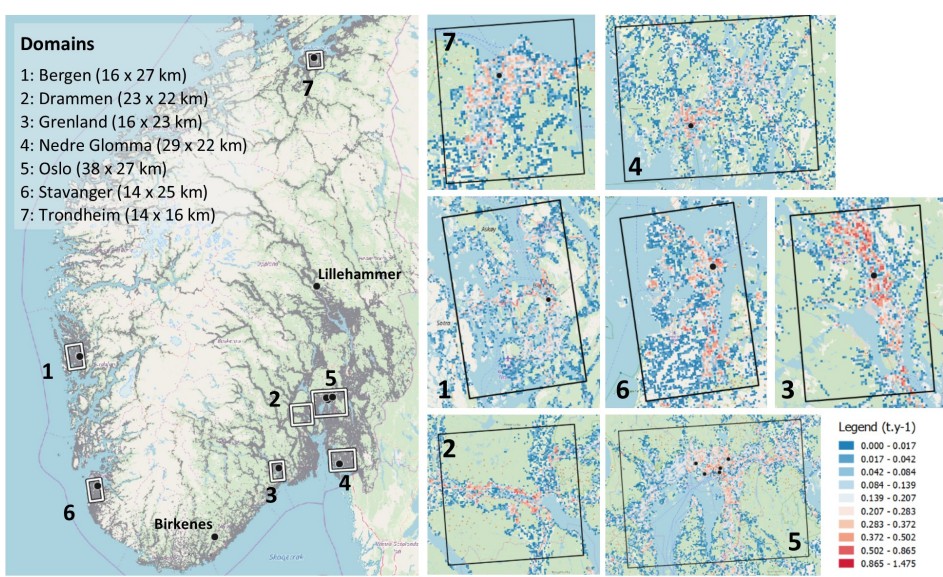

**Figure 2.** MetVed emissions $(t\,y^{-1})$ in 2015 in south of Norway and in seven urban domains at 250 m grid. The squares in the map of the south Norway represent the zoomed in domains on the right, labelled from 1 to 7 (named on the left panel), which are used for the assessment of urban emissions and dispersion modelling. The black circles represent the location of the air quality monitoring stations in Fig. 6, 7 and 8

.




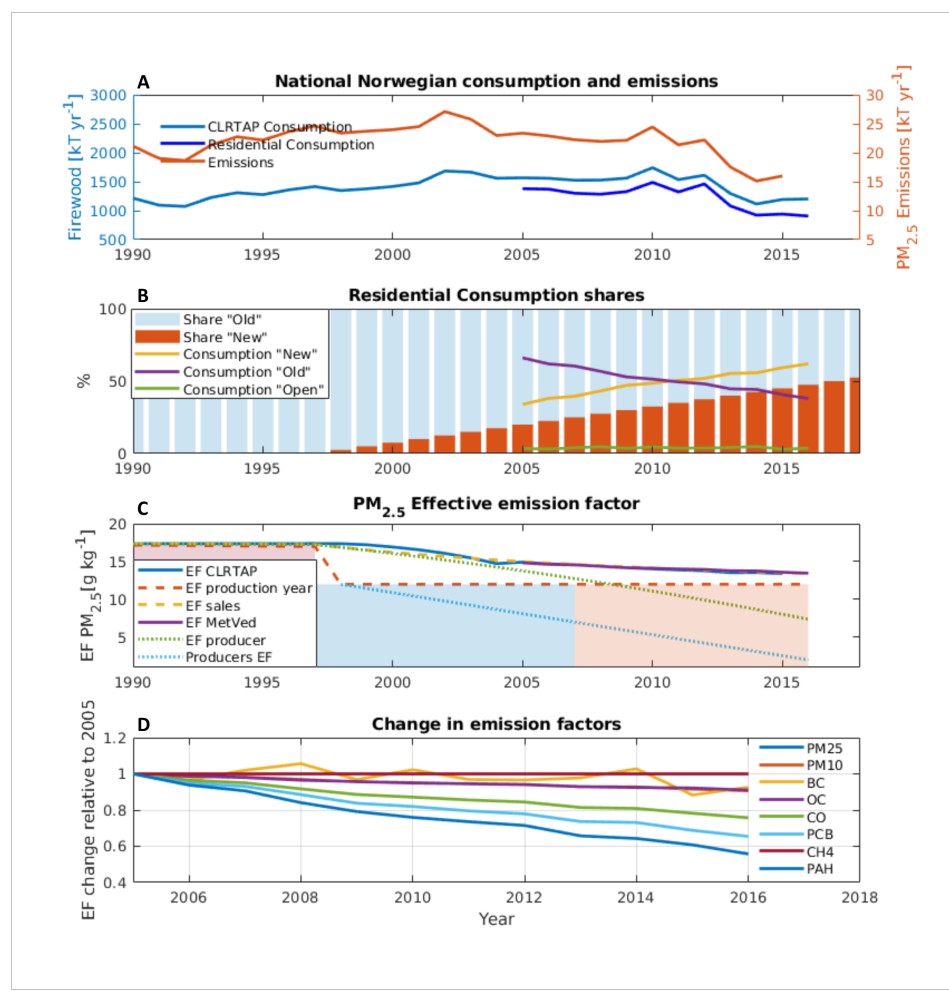

**Figure 3.** a) Historical evolution of the officially reported consumption of firewood and emissions of $PM_{2.5}$ since 1990. b) Consumption per technology class fo "New", "Old" and "Open" in yellow, blue and green, respectively. Red bars show the stock of "New" ovens assuming a constant sale over time. c) The evolution of EF, blue: derived from officially reported numbers, dark blue: MetVed emission factors, dashed red: assumed EF for oven sold in a given year and yellow dashed: assumed EF based on the sales in a). Bars show the highest allowed emissions of newly sold ovens. The dashed lines show the annual derived emission factors as reported by manufacturers (light blue) and the derived stock emission factor for each year. d) Annual average emission factors for each year based on consumption statistics.





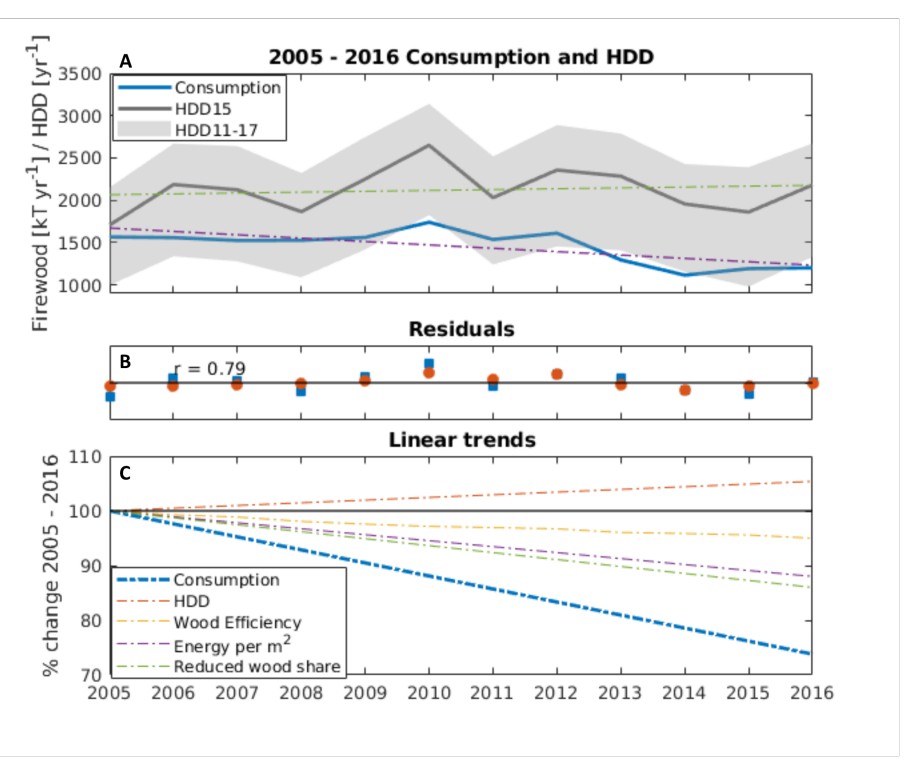

**Figure 4.** a) The HDD in the simulation domains in Fig. 2 weighted by the population in each domain along with the annually reported consumption 2005-2016. b) Residuals to linear trends 2005-2016. c) Linear trends in consumption (blue). Explanatory variables for change in consumption in the period 2005-2016, changes in heating demand (HDD), the efficiency of wood ovens (assumed 75% for "New" 50% for "Old" and 15% for "Open"), energy required to heat buildings, and wood as a share of total domestic energy consumption.





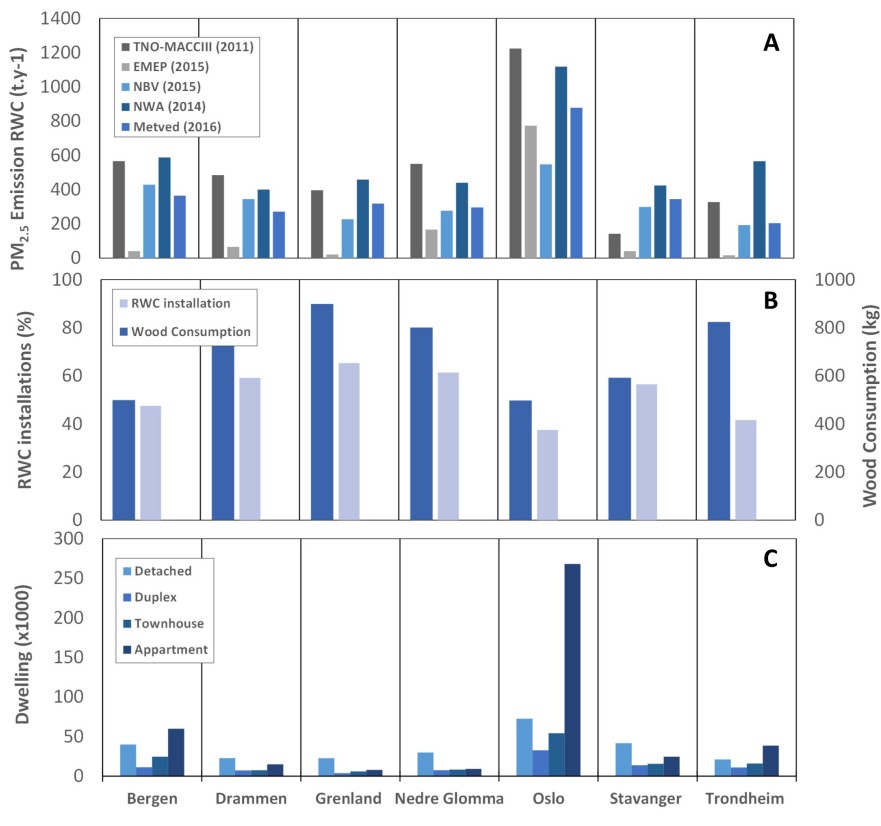

**Figure 5.** A) Total emissions within domains shown in Fig. 2. B) The % of houses with at least 1 fireplace (left y-axis) and the consumption per fireplace for each of the domains on the (right y-axis). C) The distribution of housing types for each domain.





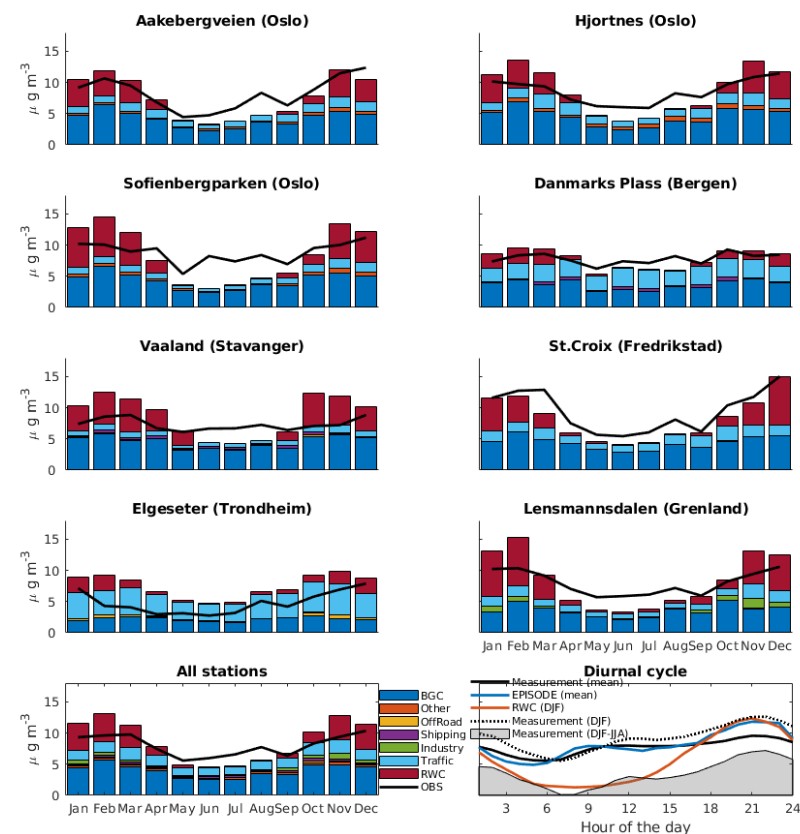

**Figure 6.** Average concentration of $PM_{2.5}$ averaged at AQ stations and as a total annual average. Bottom right: The annual average diurnal variability concentration as indicated by measurements (black) and model (blue). The orange line shows the contribution from wood burning and the shaded area is calculated as the measurement hourly average in winter (NDJF) - summer (JJA). The bars show the monthly average concentration by sector and the black line the measurement monthly average.




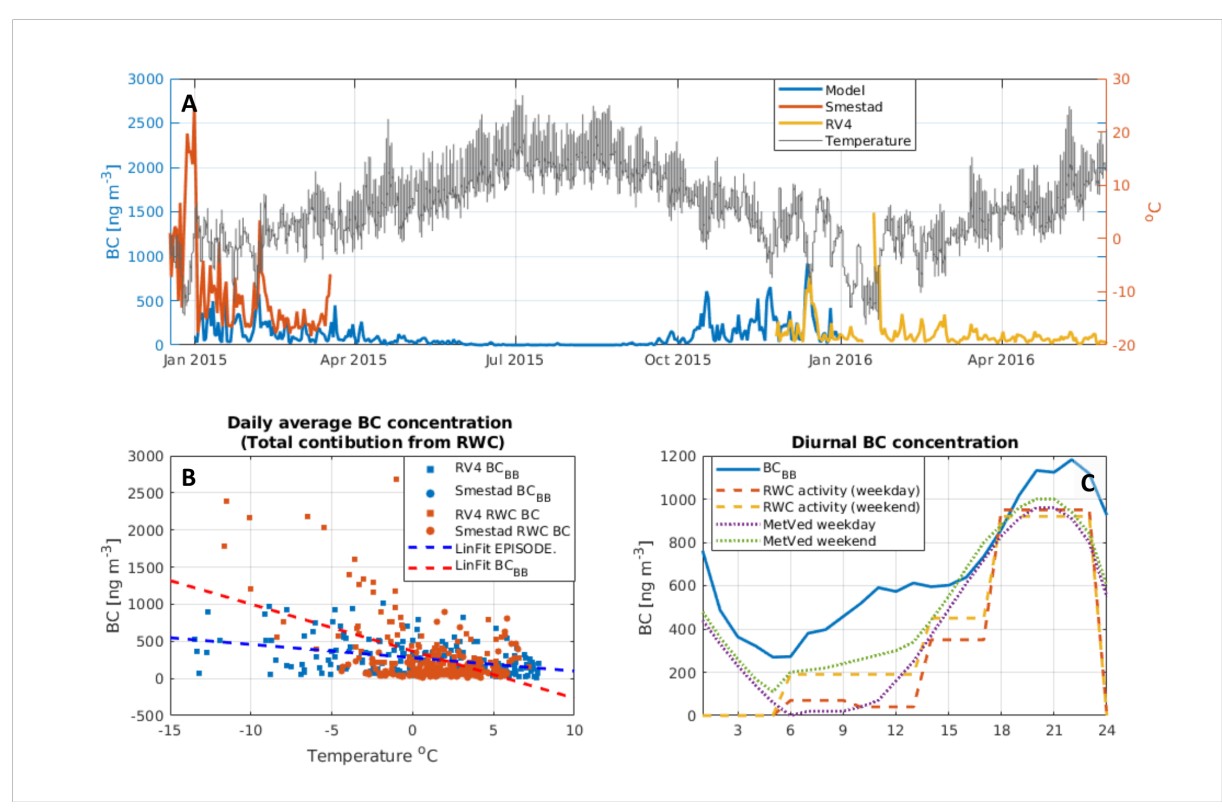

**Figure 7.** a) The left y-axis show modeled RWC BC (blue) and measured $BC_{BB}$ data at Smestad and RV4 (in red and yellow respectively). Right y-axis, timeseries of temperature at Blindern (grey) b) Aethalometer concentrations of BC from wood burning against temperature measured at two sites in Oslo winter 2015 to spring 2016 (red symbols) and EPISODE concentrations (RWC BC) in winter and spring for the calendar year of 2015 (blue symbols). c) The diurnal profile of $BC_{RWC}$ averaged over winter 2015 to spring 2016 as measured by the Aethalometer (blue). The diurnal profile of firing habits as reported by wood consumers in (Aasestad et al., 2010) shown in dashed lines for weekdays and weekends (red dashed lines). The dotted lines show the diurnal variability in emission in MetVed.





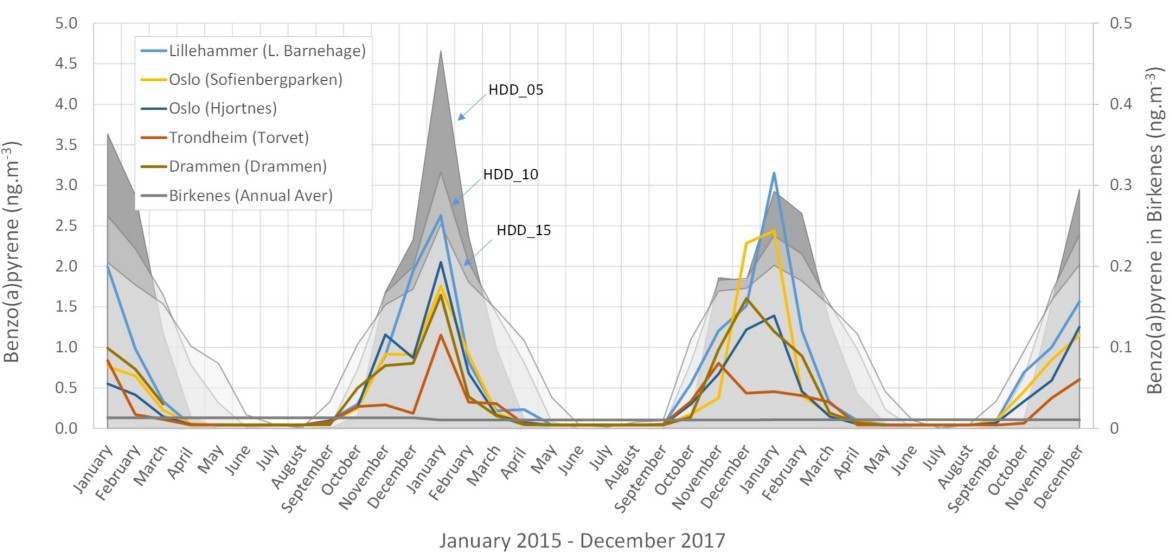

**Figure 8.** Monthly average Benzo(a)pyrene air concentrations at 5 urban sites in Norway along with the annually averaged concentration on Birkenes, a rural background station in the south of Norway. The shaded areas show the monthly RWC activity predicted by HDD with a temperature threshold of 5, 10 and 15$^o C$.