# Peer review of "The MetVed model: Development and evaluation of emissions from residential wood combustion at high spatio-temporal resolution in Norway"

_Atmospheric Chemistry and Physics, 2019_

## Referee Comment (RC1) · Anonymous Referee #2 · 19 Apr 2019

**Referee comments to acp-2019-95**

Henrik Grythe et al.: "The MetVed model: Development and evaluation of emissions from residential wood combustion at high spatio-temporal resolution in Norway"

**General comments**

The manuscript addresses an important issue regarding air quality, as RWC is a major emission source in many countries with large influence on air quality, exposure and human health. The MetVed model uses a novel approach, including different detailed data sources. It is of high importance that the methodology is applicable in a similar or adapted version for other countries, though depending on the data availability. Verification shows that the model have limitations estimating real life emissions especially in wither, but still the model provide improvements according to other models. The high temporal and spatial resolution supported by the MetVed model allow for detailed air quality modelling, exposure assessment and human health effect estimation.

The manuscript provide a novel approach, that can support and improve the temporal and spatial RWC emissions inventories not only in Norway, and is found to be a valuable input to emissions and air quality studies.

**Specific comments**

The uncertainty of wood consumption is stated to be below 3 % with reference to SSB, 2018 (p5 l3). Does the authors find this uncertainty level accurate? How is untraded fuelwood handled and how widespread is the private untraded wood for RWC?

**Technical corrections**

P2 L8 states that NOx and PM concentrations remain a major concern for human health, but health effects due to air pollution is not restricted to NOx and PM. This should be clarified.

P2 L11-12: add reference

P2 L13-14: add reference

P3 L1: "In Norway, where there are approximately 3 million individual wood burning installations, and so establishing the emissions from each individual point source constitutes a challenge" should be corrected to "In Norway, where there are approximately 3 million individual wood burning installations, establishing the emissions from each individual point source constitutes a challenge"

P3 L7-8: change "(CLRTAP; (http://ceip.at/))" to "(CLRTAP; http://ceip.at/)"

P3 L23: could this statement be supported by more references than Timmermans et al., 2013?

P4 L4: change to "…heating demand, which…"

P4 L14: change to "…in Norway, and energy…"

P9 L15-18: you introduce 3 categories (inactive, secondary and primary) of RWC installations, but you only use the latter two categories in the MetVed model as "At the moment, there is no way to establish exactly which of the listed installations are in disuse". Could you extend this part with a description of what is needed to identify inactive installations and if this will be included in an updated version of the model.

P9 L33: it would increase the clarity if the equation is extracted from the text to a separate line with equation numbering. Further, check if the equation is correct or it should be $PE_{HT}=FP_{HT}/N_{HT}$

P11 L10: missing a closing bracket

P11 L25-26: It is not clear from fig. 3d that the CH4 EF show a general decline, as stated in the text. From the figure it looks as the CH4 EF is (almost) constant.

P11 L 30-34: The difference between installations newer than 1998 is large between the fire rescue agency and the survey. Has the reason for the difference been evaluated? If the fire rescue data has large uncertainty, it is interesting to know if it is only the case for this parameter and why. If the survey have large uncertainty, e.g. due to limited number of respondents, it should be mentioned if the same survey is used for other information in the MetVed model.

P12 L4: please clarify the method for estimating sales. What is the reason for choosing this methodology and what is the data foundation?

P12 L11: clarify if CLRTAP refer to the Norwegian emissions reported to CLRTAP, as the phrasing can be misinterpreted to refer to the reporting guidelines for CLRTAP, which include EFs for more technologies than new and old.

P12 L20: add reference to figure 4.c

P13 L21-23: consider rephrasing this to make it more easy to read.

P13 L23: change "…MetVed done based…" to "…MetVed is done based…"

P13 L 29: what does NBV refer to? Include a reference.

P14 L13: change to "…Norwegian official emission factors (Tab. 1) is used."

P14 LL 16: change to "…exception. In…"

P14 L21: change to "…EMEP) closely related…"

P 14 L26: change to "…same scale, as…"

P 15 L6: change to "…Trondheim, that has the fourth highest…"

P15 L 17: change to "…Additionally, MetVed considers the dwelling size…"

P 15 L 31: change to "…particle, subject only to…"

P16 L 22: change to "…Compared to NBV emissions, which were calibrated…"

P17 L18-19: change to "When apartment emissions were emitted in the second layer, the surface concentration was reduced to 3.76 $\mu gm^{-3}$, and when smaller buildings emit in the second layer, a further reduction to 3.19 $\mu gm^{-3}$ is observed"

P 18 L 6: change to "…dependence, suggesting that…"

P18 L12: change to "…For most of the air quality stations…"

P18 L16-17: change to "…All urban measurement sites…"

P18 L34: change to "…the region, and the area…"

P19 L3: change to "…other than wood based, which represent…"

P22-24: the layout of references needs to be standardized

P22 L2: include year ( "Aasestad, K., 2010:")

P22 L9: correct name format

P22 L13: include year ("Denby, B. R., et al., 2013:")

P23 L34: correct year to 2000

P24 L 8: include year

P27: consider rearranging the maps 1-7 according to the location on the national map

P28:
Figure 3a; consider changing the chart title to "National Norwegian firewood consumption and emissions"
Figure 3c; clarify "EF producer" and "Producers EF"
Figure 3d: change the layout. Not all categories/lines are visible, and it is not possible to distinguish $PM_{2.5}$ and PAH, and CH4 and $PM_{10}$

P28: the figure text for figure 3c include errors and must be corrected. The layout of figure 3d should be improved, as different categories are visualized with very similar colors.

P29:
Figure 4a; what do the red and the green dashed lines show?
Figure 4c; it is not clear what the yellow line shows. If it is the wood ovens efficiency, it indicates that the efficiency is decreasing. That doesn't sound correct, as the new stoves are more efficient. P13 L 11-12 seem to describe that the yellow line show the decreasing fuel consumption? Please clarify both in the text and in the figure text.

P29:
Figure 4b; Y-axis % or % change (see L19-20)?
Figure text; weighted by population or number of dwellings (see L18)?

P31: The layout of figure 6a should be improved, as different categories are visualized with very similar colors. E.g. consider to decrease the number of categories (e.g. by leaving out offroad and shipping). Consider to change the order of the categories in the legend to follow the order on the chart. Figure 6 lack indication of a and b.

---

## Referee Comment (RC2) · Anonymous Referee #1 · 24 Apr 2019

The paper presents a description and evaluation of the MetVed model, a tool that allows estimating residential wood combustion emissions for Norway at high spatial and temporal resolution. The strength of MetVed is without a doubt in its ability to combine very detailed datasets that allow reducing the uncertainty in the spatio-temporal distribution of residential wood combustion emissions, which play a key role in the PM urban levels. The paper is well written and clear and a good contribution for ACP. The following comments should be taken into account before accepting the paper.

General comments The manuscript should be accompanied by a figure that illus-

trates/summarizes the general structure/workflow of the MetVed model (i.e. inputs, main functions, outputs). The amount of information used by the model is quite large, and sometimes it is difficult to follow how all this information is combined (and how the different datasets are supplemented). This figure will be very useful for the reader to understand better how the multiple data are combined to derive hourly gridded emission data.

Emission factors are established for three categories as a function of the appliances (fireplace, old stoves and new stoves). Several studies have shown that emission factors can also largely vary according to the type of wood being burned (e.g. maritime pine, eucalyptus). An example of this are the results obtained in the AIRUSE LIFE project (http://airuse.eu/wp-content/uploads/2013/11/R09_AIRUSE-Emission-factors-for-biomass-burning.pdf). Why the wood type influence is not taken into account in the MetVed model? Is it because only one type of wood is being used in Norway? Or because this information is not available? This topic should be discussed in the paper.

The model estimates emissions for several pollutants (i.e. CO, CH4, PM10, PM25., BC and PAH) but the paper does not mention other species that are also relevant in terms of air quality such as Organic Carbon (OC, only appears in Figure 3.D), NMVOC (which influence the formation of secondary organic aerosols) or NOx or climate change (CO2). Is there a specific reason for that? Are NMVOC and NOx emission from RWC considered when performing the air quality modelling exercise?

When performing the atmospheric dispersion modelling exercise, the MetVed emissions are used as input data to the EPISODE model. It is not clear how EPISODE treats the formation of secondary particles (inorganic and organic), which may have an impact in the modelled PM2.5 concentrations. Also, it is not clear which source apportionment method is used (is it a brute force approach?). Residential wood combustion emissions can contribute considerably to the atmospheric organic aerosol burden, particularly in regions with cooler climates, through both primary emissions and significant

secondary organic aerosols (e.g. https://www.nature.com/articles/srep27881). The formation of SOA due to RWC emissions should be discussed in more detail when evaluating the results.   Specific comments P4 L19-21: This sentence should be revised. The manuscript clearly states that the authors had to perform a huge work in terms of collecting all the input data required by the model (which in some cases was facilitated through personal communication and not through open data portals). In this sense, the application of the tool to another country/region may not be as versatile and transferable as stated. Similarly, it can not be say that the model can be transferred to other emission sectors. MetVed is explicitly designed to estimate emissions for residential wood combustion emissions. Other emission sources (e.g. traffic ) would require other type of input data, algorithms and workflows, that MetVed does not currently include.

P7 L23: Can the MetVed model use gridded outdoor temperature provided by a numerical weather model?

P10 L15: A citation should be added (I recommend Quayle and Diaz (1980)). Quayle RG, Diaz HF. 1980. Heating degree day data applied to residential heating energy consumption. J. Appl. Meteorol. 19 (3): 241–246.

P11 L13: This is not shown in Figure 3.a (comparison of emissions reported by CLR-TAP and estimated by MetVed)

P13 L26: Spatial and temporal distribution

P13 L27: Not all the emission inventories used for comparison are Norwergian (i.e. TNO_MACC-iii and EMEP are European emission inventories). Also, it should be stated how subdomain emissions have been derived from the original gridded inventories (e.g. for each subdomain only grid cells completely within the domain have been considered, or all the grid cells with the centroid within the domain, etc.).

P13 L29: EMEP and TNO_MACC-iii are not urban emission inventories. This adjective should be remove in all the discussion.

P14 L1: replace 0.1° by 0.1°x0.1° (and remove 7km, it is not true). Also, add a reference to the EMEP inventory.

P14 L8: Not the same spatial resolution, slightly different (i.e. 0.125°x0.0625°). Also, replace TNO-MACC by TNO_MACC-III

P14 L35: Previously it has been stated that TNO_MACC-iii downscaling is based on population density, not dwelling density.

P15 L27: Add a reference to the EPISODE model.

P15 L29: background concentrations should be replaced by boundary conditions.

P16 L3-8: This information should be moved to the previous section (5)

P16 L23: The correlation improvement is mainly occurring in 3 stations. In the other cases differences are rather small and not statistically significant.

P16 L31: Remove "for the MetVed modelled concentration".

P17 L16-19: What happens in terms of model performance when changing the emission vertical distribution? Is the overestimation observed in Oslo (Table 2) reduced? Maybe this feature (vertical allocation of emission) could be the main reason for the general overestimation of PM2.5.

P20 L1: In terms of emissions, which is the main source contributing to total BC emissions? Considering the low BC fraction used in MetVed, I would think that probably road transport is the main contributor. Then, maybe the uncertainty comes from this emission source. Also, the uncertainty can be related to the BC fraction used in the Metved model.

P20 L25-27: This sentence should be removed. See comment on P4 L19-21.

Figure 1: The text that appears in Figures1.D,E and F is not self-explanatory (should be replaced by other options such as D. Wood burning installations, E. Density of wood

[Figure]

burning installations and F. Share (%) of wood based installations. Also, the legend in Figure 1.E is not specified.

---

## Author Comment (AC1) · 27 Jun 2019

[acp, manuscript]copernicus

[Figure]

**Overview**

In this response, reviewer comments are in blue, while the response from the authors is in black. Where applicable, text from the paper that have been changed are in red. A document with tracked changes to the original manuscript has been added. We thank reviewers for insightful comments and suggested changes, which helped improve the manuscript.

The paper presents a description and evaluation of the MetVed model, a tool that allows estimating residential wood combustion emissions for Norway at high spatial and temporal resolution. The strength of MetVed is without a doubt in its ability to combine very detailed datasets that allow reducing the uncertainty in the spatio-temporal distribution of residential wood combustion emissions, which play a key role in the PM urban levels. The paper is well written and clear and a good contribution for ACP.

The following comments should be taken into account before accepting the paper.

**General comments**

The manuscript should be accompanied by a figure that illustrates/summarizes the general structure / workflow of the MetVed model (i.e. inputs, main functions, outputs). The amount of information used by the model is quite large, and sometimes it is difficult to follow how all this information is combined (and how the different datasets are supplemented). This figure will be very useful for the reader to understand better how the multiple data are combined to derive hourly gridded emission data.

We agree with the reviewer and have added a figure showing the data flow and

the various calculations done in MetVed. It is Fig. 1 in the current version of the manuscript.

Emission factors are established for three categories as a function of the appliances (fireplace, old stoves and new stoves). Several studies have shown that emission factors can also largely vary according to the type of wood being burned (e.g. maritime pine, eucalyptus). An example of this are the results obtained inthe AIRUSE LIFE project (http://airuse.eu/wp-content/uploads/2013/11/R09_AIRUSE-Emission-factors-for-biomass-burning.pdf). Why the wood type influence is not taken into account in the MetVed model? Is it because only one type of wood is being used in Norway? Or because this information is not available? This topic should be discussed in the paper.

As the reviewer points out there are several types of wood used in Norway, and emissions (and heat output) will vary with type of wood used. Large scale production of fuelwood and sales of wood (estimated 70% of consumption) is nearly exclusively birch. However, there are no detailed description of the type of wood gathered by the estimated 30% privately produced wood. This bulk of wood must be assumed to be assorted local species which certainly include but is not limited to birch.

The assumption of birch only in MetVed stems mainly from official reporting numbers and the desire to keep in line with national reported numbers, and therefore their emission factors. There are some small changes in several parts of the document, detailing this.

The model estimates emissions for several pollutants (i.e. CO, CH4, PM10, PM25.,BC and PAH) but the paper does not mention other species that are also relevant in terms of air quality such as Organic Carbon (OC, only appears in Figure 3.D), NMVOC (which influence the formation of secondary organic aerosols) or NOx or climate change (CO2). Is there a specific reason for that? Are NMVOC and NOx

emission from RWC considered when performing the air quality modelling exercise?

Generally carbon emissions from firing wood is not counted as CO2 emissions as it is considered carbon neutral. Compounds emissions are presented but given limited presence in the paper for other species than PM and BC as the focus was on validating against observations. The limitations of the dispersion model used in regards to chemistry also make challenging to model some of the species. Therefore, with spatial and temporal resolution the same, little is gained by giving too much weight to other compounds.

When performing the atmospheric dispersion modelling exercise, the MetVed emissions are used as input data to the EPISODE model. It is not clear how EPISODE treats the formation of secondary particles (inorganic and organic), which may have an impact in the modelled PM2.5 concentrations. Also, it is not clear which source apportionment method is used (is it a brute force approach?). Residential wood combustion emissions can contribute considerably to the atmospheric organic aerosol burden, particularly in regions with cooler climates, through both primary emissions and significant secondary organic aerosols (e.g. $https://www.nature.com/articles/srep27881$). The formation of SOA due to RWC emissions should be discussed in more detail when evaluating the results.

The model EPISODE in the current setup have no secondary aerosol formation or aerosol chemistry. The simplification is somewhat counterbalanced for PM as the emission factors from Seljeskog et al.,(2013) are designed so that particles are measured when the plume is cooled and diluted and so the particle mass will include significant condensed mass. With no aerosol-chemistry in EPISODE source apportionment is trivial and a straight forward decoupled ("or brute force") method can be applied.

**Specific comments**

P4 L19-21: This sentence should be revised. The manuscript clearly states that the authors had to perform a huge work in terms of collecting all the input data required by the model (which in some cases was facilitated through personal communication and not through open data portals). In this sense, the application of the tool to another country/region may not be as versatile and transferable as stated. Similarly, it can not be say that the model can be transferred to other emission sectors. MetVed is explicitly designed to estimate emissions for residential wood combustion emissions. Other emission sources (e.g. traffic ) would require other type of input data, algorithms and work flows, that MetVed does not currently include.

Rewrote the section substantially: As the above data-sets constitute the basis for the analysis of RWC in Norwegian households we provide a detailed description of each dataset in this section. The utilisation of high resolution data is important for the MetVed model to produce valuable results. The principles behind building an emission model with more bottom up principles relies heavily on gathered underlying data. Thus to achieve accurate emissions new avenuefor data gathering is an important field of development.

P7 L23: Can the MetVed model use gridded outdoor temperature provided by a numerical weather model?

Yes, we are considering this possibility in further developments. For a forecast it would require to relate consumption / emissions to temperature for a given grid. The way the model is set up the consumption per year is fixed and a posterior distribution of given consumption.

P10 L15: A citation should be added (I recommend Quayle and Diaz (1980)).

Quayle R.G., Diaz H.F., 1980. Heating degree day data applied to residential heating energyconsumption. J. Appl. Meteorol. 19 (3): 241–246.

Reference to Quale and Diaz (1980) has been added as suggested.

P11 L13: This is not shown in Figure 3.a (comparison of emissions reported by CLR-TAP and estimated by MetVed)

The correct figure reference has been added (old Fig.2 now Fig.3)

P13 L26: Spatial and temporal distribution

Added and temporal to text.

P13 L27: Not all the emission inventories used for comparison are Norwergian (i.e.TNO-MACC-iii and EMEP are European emission inventories). Also, it should be stated how subdomain emissions have been derived from the original gridded inventories (e.g. for each subdomain only grid cells completely within the domain have been considered, or all the grid cells with the centroid within the domain, etc.).

"Norwegian emission inventories" changed to "Emission inventories covering Norway"

Also added Total emissions within each domain are computed by 1st order conservative remapping emissions for each inventory to the domain (e.g. Jones 1999).

P13 L29: EMEP and TNO-MACC-iii are not urban emission inventories. This adjective should be remove in all the discussion.

changed text to emission inventories available to do urban modelling For the remainder we have kept "urban domains" as this does not speak to the intent but rather

the fact that the chosen domains cover urban areas.

P14 L1: replace 0.1 by 0.1x0.1 (and remove 7km, it is not true). Also, add a reference to the EMEP inventory.

Changed to the correct resolution as suggested by reviewer. Specifications for the EMEP emissions appear not to be publicly available and do not have a proper reference. However, they are (also gridded) openly available at the "Centre on Emission Inventories and Projections" through https://ceip.at/.

P14 L8: Not the same spatial resolution, slightly different (i.e. 0.125x0.0625). Also,replace TNO-MACC by TNO-MACC-III

Changed to the correct resolution as suggested by reviewer.

P14 L35: Previously it has been stated that TNO-MACC-iii downscaling is based on population density, not dwelling density.

Changed statement to population.

P15 L27: Add a reference to the EPISODE model.

Hamer et al. (in prep.) added as a reference.

P15 L29: background concentrations should be replaced by boundary conditions.

Text edited as suggested by reviewer.

P16 L3-8: This information should be moved to the previous section (5)

This information is specific for PM2.5 and it is our opinion that the text belongs in section 5.1 rather than the more general section 5 which also include information on the other simulations, for which this is not relevant / correct.

P16 L23: The correlation improvement is mainly occurring in 3 stations. In the other cases differences are rather small and not statistically significant.

We agree that this point was perhaps overstated in the text, and modified the statement to reflect this more moderately tends to improves correlation at most stations.

P16 L31: Remove "for the MetVed modelled concentration".

Text edited as suggested by reviewer.

P17 L16-19: What happens in terms of model performance when changing the emission vertical distribution? Is the overestimation observed in Oslo (Table 2) reduced? Maybe this feature (vertical allocation of emission) could be the main reason for the general overestimation of PM2.5.

We also had a lot of interest in the vertical distribution of emissions. Changing the vertical distribution higher up, reduces model surface concentrations and lowering it increases model surface concentration. On the whole having higher emission altitude would reduce the bias in PM2.5 concentrations. It would however act to reduce (especially) hourly correlation, due to decreased seasonality of concentrations and diurnal concentration.

While we have no solid evidence of one emissions being too high or low, we went

with altitudes that seemed logical considering average building height of different types adding typical plume rise. We have not succeeded finding research material well documenting what the "correct" altitude "should" but have argued that separating apartment from others is sensible and produces improved results in our domains.

P20 L1: In terms of emissions, which is the main source contributing to total BC emissions? Considering the low BC fraction used in MetVed, I would think that probably road transport is the main contributor. Then, maybe the uncertainty comes from this emission source. Also, the uncertainty can be related to the BC fraction used in the Metved model.

As these are road near measurements, the report by C. Hak (2017) shows less than 10 % biomass burning derived BC, the remaining predominantly from traffic (fossile). We feel the distinctive diurnal and seasonal cycle makes a strong case for the separation, supporting the instruments partitioning of the two. Though we accept your point that a small partitioning error of traffic concentrations would lead to large concentration differences of $BC_{BB}$. We agree that there are a number of sources of uncertainty, but believe more in the latter cause you give (BC fraction).

P20 L25-27: This sentence should be removed. See comment on P4

we modified the text here in accordance with comments on P4.

L19-21. Figure 1: The text that appears in Figures 1.D,E and F is not self-explanatory (should be replaced by other options such as D. Wood burning installations, E. Density of woodburning installations and F. Share (%) of wood based installations. Also, the legend in Figure 1.E is not specified.

We have followed the reviewers recommendations and redone the figure and

text.

**References**

Hak, C.: Vurdering av Black Carbon (BC) og CO2 langs veg i Oslo. NILU/OR Report 11/2017. Norwegian Institute for Air Research, Kjeller, Norway. (In Norwegian), 2017.

Jones, P. W.: First- and second-order conservative remapping schemes for grids in spherical coordinates. Mon. Wea. Rev., 127, 2204–2210. doi: 10.1175/1520-0493(1999), 1999.

---

## Author Response (AR1)

https://www.overleaf.com/project/59c0f3091cf98f2fe3a3cade

[revised manuscript text omitted]

**Overview**

In this response, reviewer comments are in blue, while the response from the authors is in black. Where applicable, text from the paper that have been changed are in red. A document with tracked changes to the original manuscript has been added. We thank reviewers for insightful comments and suggested changes, which helped improve the manuscript.

[Figure]

**Figure 1.** Schematic representation of the data flow and calculations done in MetVed.

The paper presents a description and evaluation of the MetVed model, a tool that allows estimating residential wood combustion emissions for Norway at high spatial and temporal resolution. The strength of MetVed is without a doubt in its ability to combine very detailed datasets that allow reducing the uncertainty in the spatio-temporal distribution of residential wood combustion emissions, which play a key role in the PM urban levels. The paper is well written and clear and a good contribution for ACP.

The following comments should be taken into account before accepting the paper.

**General comments**

The manuscript should be accompanied by a figure that illustrates/summarizes the general structure / workflow of the MetVed model (i.e. inputs,main functions, outputs). The amount of information used by the model is quite large, and sometimes it is difficult to follow how all this information is combined (and how the different datasets are supplemented). This figure will be very useful for the reader to understand better how the multiple data are combined to derive hourly gridded emission data.

We agree with the reviewer and have added a figure showing the data flow and the various calculations done in MetVed. It is Fig. 1 in the current version of the manuscript.

Emission factors are established for three categories as a function of the appliances (fireplace, old stoves and new stoves).

[Figure]

**Figure 2.** Example of part of the input data used in MetVed model in the Municipality of Stavanger. A: Total dwelling number at 250 m grid resolution. B: Number of detached hours. C: Number of apartments. D: Individual wood burning installations from the Fire and Rescue Agency. E: Density of wood burning installations. F: Share (%) of wood based installations for residential heating obtained from the webcrawled data-set.

Several studies have shown that emission factors can also largely vary according to the type of wood being burned (e.g. maritime pine, eucalyptus). An example of this are the results obtained inthe AIRUSE LIFE project (http://airuse.eu/wp-content/uploads/2013/11/R09_AIRUSE-Emission-factors-for-biomass-burning.pdf). Why the wood type influence is not taken into account in the MetVed model? Is it because only one type of wood is being used in Norway? Or because this information is not available? This topic should be discussed in the paper.

As the reviewer points out there are several types of wood used in Norway, and emissions (and heat output) will vary with type of wood used. Large scale production of fuelwood and sales of wood (estimated 70% of consumption) is nearly exclu-

[Figure]

**Figure 3.** MetVed emissions ($\mathrm{t\,y^{-1}}$) in 2015 in south of Norway and in seven urban domains at 250 m grid. The squares in the map of the south Norway represent the zoomed in domains on the right, labelled from 1 to 7 (named on the left panel), which are used for the assessment of urban emissions and dispersion modelling. The black circles represent the location of the air quality monitoring stations in Fig. 7, 8 and 9

.

sively birch. However, there are no detailed description of the type of wood gathered by the estimated 30% privately produced wood. This bulk of wood must be assumed to be assorted local species which certainly include but is not limited to birch.

The assumption of birch only in MetVed stems mainly from official reporting numbers and the desire to keep in line with national reported numbers, and therefore their emission factors. There are some small changes in several parts of the document, detailing this.

The model estimates emissions for several pollutants (i.e. CO, CH4, PM10, PM25.,BC and PAH) but the paper does not mention other species that are also relevant in terms of air quality such as Organic Carbon (OC, only appears in Figure 3.D), NMVOC (which influence the formation of secondary organic aerosols) or NOx or climate change (CO2). Is there a specific reason for that? Are NMVOC and NOx emission from RWC considered when performing the air quality modelling exercise?

Generally carbon emissions from firing wood is not counted as CO2 emissions as it is considered carbon neutral. Compounds emissions are presented but given limited presence in the paper for other species than PM and BC as the focus was on validating against observations. The limitations of the dispersion model used in regards to chemistry also make challenging to model some of the species. Therefore, with spatial and temporal resolution the same, little is gained by giving too much weight to other compounds.

[Figure]

**Figure 4.** a) Historical evolution of the officially reported consumption of firewood and emissions of $PM_{2.5}$ since 1990. b) Consumption per technology class  for "New", "Old" and "Open" in yellow, blue and green, respectively. Red bars show the stock of "New" ovens assuming a constant sale over time. c) The evolution of  the average emission factors over time. The emission factor derived from officially reported numbers (red),  MetVed emission factors  (blue). Yellow dashed  line show emission factor on the oven sales (red bars in a)  and equal consumption in all ovens. The  dotted dark blue  line: the derived  annual emission factor when manufacturer information on emission factor are used for each year. d) Annual average emission factors for each year of each component in the MetVed emissions, based on consumption statistics.

When performing the atmospheric dispersion modelling exercise, the MetVed emissions are used as input data to the EPISODE model. It is not clear how EPISODE treats the formation of secondary particles (inorganic and organic), which may have an impact in the modelled PM2.5 concentrations. Also, it is not clear which source apportionment method is used (is it a brute force approach?). Residential wood combustion emissions can contribute considerably to the atmospheric organic aerosol burden, particularly in regions with cooler climates, through both primary emissions and significant secondary organic aerosols

[Figure]

**Figure 5.** a) The HDD in the simulation domains in Fig. 3 weighted by the population in each domain along with the annually reported consumption 2005-2016. Dashed line is the linear trend. b)  Normalized residuals to linear trends 2005-2016. c)  The blue line show the linear trend in consumption. Explanatory variables for this change  (2005-2016) are, changes in heating demand (HDD "COLOR!"), in "COLOR!" the efficiency of wood ovens (assumed 75% for "New" 50% for "Old" and 15% for "Open"), in "COLOR!", the reduced energy required to heat buildings, and  the decreasing share of total domestic energy consumption from wood "COLOR!".

(e.g. $https://www.nature.com/articles/srep27881$). The formation of SOA due to RWC emissions should be discussed in more detail when evaluating the results.

The model EPISODE in the current setup have no secondary aerosol formation or aerosol chemistry. The simplification is

5  somewhat counterbalanced for PM as the emission factors from Seljeskog et al.,(2013) are designed so that particles are measured when the plume is cooled and diluted and so the particle mass will include significant condensed mass. With no aerosol-chemistry in EPISODE source apportionment is trivial and a straight forward decoupled ("or brute force") method can be applied.

[Figure]

**Figure 6.** A) Total emissions within domains shown in Fig. 3. B) The % of houses with at least 1 fireplace (left y-axis) and the consumption per fireplace for each of the domains on the (right y-axis). C) The distribution of housing types for each domain.

**Specific comments**

P4 L19-21: This sentence should be revised. The manuscript clearly states that the authors had to perform a huge work in terms of collecting all the input data required by the model (which in some cases was facilitated through personal communication and not through open data portals). In this sense, the application of the tool to another country/region may not be as versatile and transferable as stated. Similarly, it can not be say that the model can be transferred to other emission sectors. MetVed is explicitly designed to estimate emissions for residential wood combustion emissions. Other emission sources (e.g. traffic ) would require other type of input data, algorithms and work flows, that MetVed does not currently include.

Rewrote the section substantially: As the above data-sets constitute the basis for the analysis of RWC in Norwegian households we provide a detailed description of each dataset in this section. The utilisation of high resolution data is important for the MetVed model to produce valuable results. The principles behind building an emission model with more bottom up principles relies heavily on gathered underlying data. Thus to achieve accurate emissions new avenuefor data gathering is an important

[Figure]

**Figure 7.** Average concentration of $PM_{2.5}$ averaged at AQ stations and as a total annual average. Bottom right: The annual average diurnal variability concentration as indicated by measurements (black) and model (blue). The orange line shows the contribution from wood burning and the shaded area is calculated as the measurement hourly average in winter (NDJF) - summer (JJA). The bars show the monthly average concentration by sector and the black line the measurement monthly average.

field of development.

P7 L23: Can the MetVed model use gridded outdoor temperature provided by a numerical weather model?

5   Yes, we are considering this possibility in further developments. For a forecast it would require to relate consumption / emissions to temperature for a given grid. The way the model is set up the consumption per year is fixed and a posterior distribution of given consumption.

[Figure]

**Figure 8.** a) The left y-axis show modeled RWC BC (blue) and measured $BC_{BB}$ data at Smestad and RV4 (in red and yellow respectively). Right y-axis, timeseries of temperature at Blindern (grey) b) Aethalometer concentrations of BC from wood burning against temperature measured at two sites in Oslo winter 2015 to spring 2016 (red symbols) and EPISODE concentrations (RWC BC) in winter and spring for the calendar year of 2015 (blue symbols). c) The diurnal profile of $BC_{RWC}$ averaged over winter 2015 to spring 2016 as measured by the Aethalometer (blue). The diurnal profile of firing habits as reported by wood consumers in (Aasestad et al., 2010) shown in dashed lines for weekdays and weekends (red dashed lines). The dotted lines show the diurnal variability in emission in MetVed.

P10 L15: A citation should be added (I recommend Quayle and Diaz (1980)). Quayle R.G., Diaz H.F., 1980. Heating degree day data applied to residential heating energyconsumption. J. Appl. Meteorol. 19 (3): 241–246.

Reference to Quale and Diaz (1980) has been added as suggested.

P11 L13: This is not shown in Figure 3.a (comparison of emissions reported by CLR-TAP and estimated by MetVed)

The correct figure reference has been added (old Fig.2 now Fig.3)

10  P13 L26: Spatial and temporal distribution

[Figure]

**Figure 9.** Monthly average Benzo(a)pyrene air concentrations at 5 urban sites in Norway along with the annually averaged concentration on Birkenes, a rural background station in the south of Norway. The shaded areas show the monthly RWC activity predicted by HDD with a temperature threshold of 5, 10 and 15$^o$C.

Added and temporal to text.

P13 L27: Not all the emission inventories used for comparison are Norwergian (i.e.TNO-MACC-iii and EMEP are European emission inventories). Also, it should be stated how subdomain emissions have been derived from the original gridded inventories (e.g. for each subdomain only grid cells completely within the domain have been considered, or all the grid cells with the centroid within the domain, etc.).

"Norwegian emission inventories" changed to "Emission inventories covering Norway"

Also added Total emissions within each domain are computed by 1st order conservative remapping emissions for each inventory to the domain (e.g. Jones 1999).

P13 L29: EMEP and TNO-MACC-iii are not urban emission inventories. This adjective should be remove in all the discussion.

changed text to emission inventories available to do urban modelling For the remainder we have kept "urban domains" as this does not speak to the intent but rather the fact that the chosen domains cover urban areas.

P14 L1: replace 0.1 by 0.1x0.1 (and remove 7km, it is not true). Also, add a reference to the EMEP inventory.

Changed to the correct resolution as suggested by reviewer. Specifications for the EMEP emissions appear not to be publicly

5  available and do not have a proper reference. However, they are (also gridded) openly available at the "Centre on Emission Inventories and Projections" through https://ceip.at/.

P14 L8: Not the same spatial resolution, slightly different (i.e. 0.125x0.0625). Also,replace TNO-MACC by TNO-MACC-III

Changed to the correct resolution as suggested by reviewer.

P14 L35: Previously it has been stated that TNO-MACC-iii downscaling is based on population density, not dwelling density.

Changed statement to population.

P15 L27: Add a reference to the EPISODE model.

20  Hamer et al. (in prep.) added as a reference.

P15 L29: background concentrations should be replaced by boundary conditions.

Text edited as suggested by reviewer.

P16 L3-8: This information should be moved to the previous section (5)

This information is specific for PM2.5 and it is our opinion that the text belongs in section 5.1 rather than the more general section 5 which also include information on the other simulations, for which this is not relevant / correct.

P16 L23: The correlation improvement is mainly occurring in 3 stations. In the other cases differences are rather small and not statistically significant.

We agree that this point was perhaps overstated in the text, and modified the statement to reflect this more moderately tends to

35  improves correlation at most stations.

P16 L31: Remove "for the MetVed modelled concentration".

Text edited as suggested by reviewer.

P17 L16-19: What happens in terms of model performance when changing the emission vertical distribution? Is the overestimation observed in Oslo (Table 2) reduced? Maybe this feature (vertical allocation of emission) could be the main reason for the general overestimation of PM2.5.

10 We also had a lot of interest in the vertical distribution of emissions. Changing the vertical distribution higher up, reduces model surface concentrations and lowering it increases model surface concentration. On the whole having higher emission altitude would reduce the bias in PM2.5 concentrations. It would however act to reduce (especially) hourly correlation, due to decreased seasonality of concentrations and diurnal concentration.

While we have no solid evidence of one emissions being too high or low, we went with altitudes that seemed logical con-
15 sidering average building height of different types adding typical plume rise. We have not succeeded finding research material well documenting what the "correct" altitude "should" but have argued that separating apartment from others is sensible and produces improved results in our domains.

P20 L1: In terms of emissions, which is the main source contributing to total BC emissions? Considering the low BC fraction
used in MetVed, I would think that probably road transport is the main contributor. Then, maybe the uncertainty comes from this emission source. Also, the uncertainty can be related to the BC fraction used in the Metved model.

As these are road near measurements, the report by C. Hak (2017) shows less than 10 % biomass burning derived BC, the remaining predominantly from traffic (fossile). We feel the distinctive diurnal and seasonal cycle makes a strong case for the
25 separation, supporting the instruments partitioning of the two. Though we accept your point that a small partitioning error of traffic concentrations would lead to large concentration differences of $BC_{BB}$. We agree that there are a number of sources of uncertainty, but believe more in the latter cause you give (BC fraction).

P20 L25-27: This sentence should be removed. See comment on P4

we modified the text here in accordance with comments on P4.

L19-21. Figure 1: The text that appears in Figures 1.D,E and F is not self-explanatory (should be replaced by other options such as D. Wood burning installations, E. Density of woodburning installations and F. Share (%) of wood based installations.
35 Also, the legend in Figure 1.E is not specified.

We have followed the reviewers recommendations and redone the figure and text.

**10 General comments**

In this response, rewiever comments are in blue, while the author response is in black. Where applicable, text from the paper that have been changed are in red. A document with tracked changes to the original manuscript has been added. We thank reviewers for insightful comments and suggested changes.

The manuscript addresses an important issue regarding air quality, as RWC is a major emission source in many countries with large influence on air quality, exposure and human health. The MetVed model uses a novel approach, including different detailed data sources. It is of high importance that the methodology is applicable in a similar or adapted version for other countries, though depending on the data availability.Verification shows that the model have limitations estimating real life emissions especially in wither, but still the model provide improvements according to other models. The high temporal and spatial resolution supported by the MetVed model allow for detailed air quality modelling, exposure assessment and human health effect estimation.

The manuscript provide a novel approach, that can support and improve the temporal and spatial RWC emissions inventories not only in Norway, and is found to be a valuable input to emissions and air quality studies.

**25 Specific comments**

The uncertainty of wood consumption is stated to be below 3 % with reference to SSB, 2018 (p5 l3). Does the authors find this uncertainty level accurate? How is untraded fuelwood handled and how widespread is the private untraded wood for RWC?

The total uncertainty on Norwegian consumption is larger than the 3% given in the text, this is the sample size uncertainty regarding representativeness of the interviewees. In Norway the consumption estimates are not based on sales, but on telephone interviews. The sample size uncertainty is 3% for Norway as a whole, which does not include uncertainty in peoples memory, uncertainty regarding quantification by different people etc. As the measurement is on consumption and not sales related, thus untraded fuels do not represent additional fuel or uncertainty in this regard.

**Technical corrections**

P2 L8 states that NOx and PM concentrations remain a major concern for human health, but health effects due to air pollution is not restricted to NOx and PM. This should be clarified.

10  added preposition "among" to sentence, it now reads: Together with nitrogen oxides (NOx), elevated particulate matter (PM) concentrations remain among the major concerns for human health.

P2 L11-12: add reference

15  Reference to Genberg et al.,(2011) has been added.

P2 L13-14: add reference

Reference to Karagulian et al.,(2015) has been added.

P3 L1: "In Norway, where there are approximately 3 million individual wood burning installations, and so establishing the emissions from each individual point source constitutes a challenge" should be corrected to "In Norway, where there are approximately 3 million individual wood burning installations, establishing the emissions from each individual point source constitutes a challenge"

Text changed as suggested by reviewer.

P3 L7-8: change "(CLRTAP; (http://ceip.at/))" to "(CLRTAP; http://ceip.at/)"

30  Text changed as suggested by reviewer.

P3 L23: could this statement be supported by more references than Timmermans et al., 2013?

Text changed as suggested by reviewer.

Text changed as suggested by reviewer.

P9 L15-18: you introduce 3 categories (inactive, secondary and primary) of RWC installations, but you only use the latter two categories in the MetVed model as "At the moment, there is no way to establish exactly which of the listed installations are in disuse". Could you extend this part with a description of what is needed to identify inactive installations and if this will be included in an updated version of the model.

Added section: To detail out further the activity level of fireplaces, further data could be collected. In some municipalities, records of the residue material from chimneys swept are kept record of and graded on a scale from 1-9 (clean to dirty). These and similar data could be used directly to estimate the activity in each chimney, but would need a proper framework. Also consumption questionnaire presently asks respondents first if they have a wood firing installation then if it is in use, so this also supports finding the average share of inactive fireplaces.

P9 L33: it would increase the clarity if the equation is extracted from the text to a separate line with equation numbering. Further, check if the equation is correct or it should be PEHT=FPHT/NHT

The equation is extracted to its own line. The equation is correct as it stands.

inserted.

P11 L25-26: It is not clear from fig. 3d that the CH4 EF show a general decline, as stated in the text. From the figure it looks as the CH4 EF is (almost) constant.

Both figure 3 and 4 together with the text describing them has been redone to be clearer.

As the reviewer points out, the methane emission factor is constant over time as the emission factor for "New" and "Old" ovens are the same. inserted except $CH_4$

 The difference between installations newer than 1998 is large between the fire rescue agency and the survey. Has the reason for the difference been evaluated? If the fire rescue data has large uncertainty, it is interesting to know if it is only the case for this parameter and why. If the survey have large uncertainty, e.g. due to limited number of respondents, it should be mentioned if the same survey is used for other information in the MetVed model.

In section 2.4 we write in detail on this. It is an interesting difference for a number of reasons. Though it is hard to conclude there are good reasons to believe the fire department underestimate the fraction of new ovens. In addition there are good reasons to assume that the average consumption is higher in newer ovens. The implications of this is that the role of the differentiating done from fire departments are used as statistical input (see sec. 3.4).

P12 L4: please clarify the method for estimating sales. What is the reason for choosing this methodology and what is the data foundation?

It is clarified ion the text that these sales numbers are based on the survey in the previous sentence.

P12 L11: clarify if CLRTAP refer to the Norwegian emissions reported to CLRTAP, as the phrasing can be misinterpreted to refer to the reporting guidelines for CLRTAP, which include EFs for more technologies than new and old.

Sentence now reads: Both Norwegian emissions reported to CLRTAP and MetVed assumes constant EF for the "New" oven assembly.

P12 L20: add reference to figure 4.c

added figure reference.

P13 L21-23: consider rephrasing this to make it more easy to read.

It now reads Due to large differences in the input data, the housing type and size and energy dependencies calculated within MetVed is done only based on the ENOVA reported total energy consumption.

P13 L23: change "...MetVed done based..." to "...MetVed is done based..."

Text changed as suggested by reviewer.

P13 L 29: what does NBV refer to? Include a reference.

5  ”Norsk Beregningsverktøy” added to text

P14 L13: change to “...Norwegian official emission factors (Tab. 1) is used.”

Text changed as suggested by reviewer.

P14 L16: change to “...exception. In...”

Text changed as suggested by reviewer.

15  P14 L21: change to “...EMEP) closely related...”

Not sure what the change would be.

P 14 L26: change to “...same scale, as...”

Text changed as suggested by reviewer.

P 15 L6: change to “...Trondheim, that has the fourth highest...”

25  Text changed as suggested by reviewer.

P15 L 17: change to “...Additionally, MetVed considers the dwelling size...”

Text changed as suggested by reviewer.

P 15 L 31: change to “...particle, subject only to...”

Text changed as suggested by reviewer.

35  P16 L 22: change to “...Compared to NBV emissions, which were calibrated...”

Text changed as suggested by reviewer.

P17 L18-19: change to "When apartment emissions were emitted in the second layer,the surface concentration was reduced to 3.76 µg$^{-3}$, and when smaller buildings emit in the second layer, a further reduction to 3.19 µg$^{-3}$ is observed"

Text changed as suggested by reviewer.

P 18 L 6: change to "...dependence, suggesting that..."

Text changed as suggested by reviewer.

P18 L12: change to "...For most of the air quality stations..."

Text changed as suggested by reviewer.

P18 L16-17: change to "...All urban measurement sites..."

Text changed as suggested by reviewer.

P18 L34: change to "...the region, and the area..."

Text changed as suggested by reviewer.

P19 L3: change to "...other than wood based, which represent..."

Text changed as suggested by reviewer.

P22-24: the layout of references needs to be standardised

Journal article references have been standardised.

P22 L2: include year ("Aasestad, K., 2010:")

Journal article references have been standardised.

P22 L9: correct name format

Journal article references have been standardised.

P22 L13: include year("Denby, B. R., et al., 2013:")

Journal article references have been standardised.

P23 L34: correct year to 2000

GAINS reference corrected.

P24 L 8: include year

Seljeskog reference, year added.

P27: consider rearranging the maps 1-7 according to the location on the national map

This did not look good due to domain sizes and was not done.

P28:Figure 3a; consider changing the chart title to "National Norwegian firewood consumption and emissions" Figure 3c; clarify "EF producer" and "Producers EF" Figure 3d: change the layout. Not all categories/lines are visible, and it is not possible to distinguish PM2.5 and PAH, and CH4 and PM10

Both figure 3 and 4 together with the text describing them has been redone to be clearer.

P28: the figure text for figure 3c include errors and must be corrected. The layout of figure 3d should be improved, as different categories are visualized with very similar colors.

Both figure 3 and 4 together with the text describing them has been redone to be clearer.

P29: Figure 4a; what do the red and the green dashed lines show? Figure 4c; it is not clear what the yellow line shows. If it is the wood ovens efficiency, it indicates that the efficiency is decreasing. That doesn't sound correct, as the new stoves are more efficient.

Both figure 3 and 4 together with the text describing them has been redone to be clearer.

P13 L 11-12 seem to describe that the yellow line show the decreasing fuel consumption? Please clarify both in the text and in the figure text.

We have slightly rewritten the paragraph on page 13 to be more clear. Both figure 3 and 4 together with the text describing them has been redone to be clearer.

P29: Figure 4b; Y-axis % or % change (see L19-20)? Figure text; weighted by population or number of dwellings (see L18)?

Both figure 3 and 4 together with the text describing them has been redone to be clearer.

4c, You are absolutely right, this is not % change, but % of 2005 demand. In 4b the residuals are normalised and thus unit less as shown.

P31: The layout of figure 6a should be improved, as different categories are visualized with very similar colors. E.g. consider to decrease the number of categories (e.g. by leaving out offroad and shipping). Consider to change the order of the categories in the legend to follow the order on the chart.Figure 6 lack indication of a and b.

We decided not to remove emission sectors from this figure. While not all domains have i.e. industrial emissions and not all stations are influenced significantly by emissions in each domain, the sectors overlap those used in other Norwegian AQ modelling and to connect it to other work done on these stations we feel that the loss of some layout is compensated by gains in the information the figure gives.